

# Measured solid state and sub-cooled liquid vapour pressures of nitroaromatics using Knudsen effusion mass spectrometry

Petroc D. Shelley[1], Thomas J. Bannan[1], Stephen D. Worrall[2], M. Rami Alfarra[1,3], Ulrich K. Krieger[4], Carl J. Percival[5], Arthur Garforth[6], David Topping[1]

[1]Department of Earth and Environmental Sciences, University of Manchester, Manchester, UK
[2]School of Engineering and Applied Science, Aston University, Birmingham, UK
[3]National Centre for Atmospheric Science (NCAS), University of Manchester, Manchester, UK
[4]Institute for Atmospheric and Climate Science, ETH Zurich, Switzerland
[5]Jet Propulsion Laboratory, Pasadena, CA 91109, USA
[6]Department of Chemical Engineering & Analytical Science, University of Manchester, Manchester, UK

Correspondence to: Petroc D. Shelley (petroc.shelley@manchester.ac.uk), Thomas J. Bannan (Thomas.bannan@manchester.ac.uk), Stephen D. Worrall (s.worrall@aston.ac.uk), M. Rami Alfarra (rami.alfarra@manchester.ac.uk), Ulrich K. Krieger (ulrich.krieger@env.ethz.ch), Carl J. Percival (carl.j.percival@jpl.nasa.gov), Arthur Garforth (arthur.garforth@manchester.ac.uk), David Topping (david.topping@manchester.ac.uk)

**Abstract.** Knudsen Effusion Mass Spectrometry (KEMS) was used to measure the solid state saturation vapour pressure ($P_S^{sat}$) of a range of atmospherically relevant nitroaromatic compounds over the temperature range from 298 to 328 K. The selection of species analysed contained a range of geometric isomers and differing functionalities, allowing for the impacts of these factors on saturation vapour pressure ($P^{sat}$) to be probed. Three subsets of nitroaromatics were investigated, nitrophenols, nitrobenzaldehydes and nitrobenzoic acids. The $P_S^{sat}$ were converted to sub-cooled liquid saturation vapour pressures ($P_L^{sat}$) using experimental enthalpy of fusion and melting point values measured using differential scanning calorimetry (DSC). The $P_L^{sat}$ were compared to those estimated by predictive techniques and, with a few exceptions, were found to be up to 7 orders of magnitude lower. The large differences between the estimated $P_L^{sat}$ and the experimental can be attributed to the predictive techniques not containing parameters to adequately account for functional group positioning around an aromatic ring, or the interactions between said groups. When comparing the experimental $P_S^{sat}$ of the measured compounds the ability to hydrogen bond (H-Bond), and the strength of a H-bond formed appear to have the strongest influence on the magnitude of the $P^{sat}$ with steric effects and molecular weight also being major factors. Comparisons were made between the KEMS system and data from diffusion-controlled evaporation rates of single particles in an electrodynamic balance (EDB). The KEMS and the EDB showed good agreement with each other for the compounds investigated.





## 1 Introduction

Organic aerosols (OA) are an important component of the atmosphere with regards to resolving the impact aerosols have on both climate and air quality (Kroll and Seinfeld, 2008). To predict how OA will behave requires knowledge of their physiochemical properties. OA consist of primary organic aerosols (POA) and secondary organic aerosols (SOA). POA are

emitted directly into the atmosphere as solid or liquid particulates and make up about 20% of OA mass globally (Ervens et al., 2011), but the exact percentage of POA varies by a significant amount from region to region. SOA are not emitted into the atmosphere directly as aerosols, but instead form through atmospheric processes such as gas phase photochemical reactions (Ervens et al., 2011) or gas-to-particle conversion in the atmosphere (Pöschl, 2005). A key property for predicting the partitioning of compounds between the gaseous and aerosol phase is the pure component equilibrium vapour pressure,

also known as the saturation vapour pressure ($P^{sat}$) (Bilde et al., 2015). It has been estimated that the number of organic compounds in the atmosphere is in excess of 100,000 (Hallquist et al., 2009); therefore it is not feasible to measure the $P^{sat}$ of each experimentally. Instead, $P^{sat}$ are often estimated using group contribution methods (GCMs) that are designed to capture the functional dependencies on predicting absolute values. GCMs start with a base molecule with known properties, typically the carbon skeleton. A functional group is then added to the base molecule. This addition will change

the $P^{sat}$ and the difference between the base molecule and the functionalised molecule is the contribution from that particular functional group. If this concept is true then the contribution from the functional group should not be affected by the base molecule to which it is added (Bilde et al., 2015). Whilst this is true in many cases, there are numerous exceptions. These exceptions normally occur when proximity effects occur, such as neighbouring group interactions or other mesomeric effects. In this work there will be a focus on the Nannoolal et al. method (Nannoolal et al., 2008), the Myrdal and Yalkowsky

method (Myrdal and Yalkowsky, 1997), SIMPOL (Pankow and Asher, 2008) and EVAPORATION (Compernolle et al., 2011). Detailed assessments of such methods have been made by Barley and McFiggans (Barley and McFiggans, 2010) and O'Meara et al. (O 'meara et al., 2014) often showing predicted values differ significantly from experimental data. The limitations and uncertainties of GCMs come from a range of factors including underrepresentation of long chain hydrocarbons ($>C_{18}$), underrepresentation of certain functional groups, such as nitro or nitrate groups, a lack of data for the

impact of intramolecular bonding, and the temperature dependence due to the need for extrapolation over large temperature



ranges to reach ambient conditions (Bilde et al., 2015). This has important implications for partitioning modelling, in a mechanistic sense, such as an over or underestimation of the fraction partitioning to the particulate state. Different GCMs have different levels of reliability for different classes of compound and perform much more reliably if the compound of interest resembles those used in the parametrisation data set of the GCM (Kurtén et al., 2016). For example, it has been

recommend that the EVAPORATION method should not be used for aromatic compounds, as there are no aromatic compounds in the parametrisation dataset (Compernolle et al., 2011). However, even the methods developed with OA in mind, such as the EVAPORATION method, are not without their limitations due to the lack of experimental data available for highly functionalised, low volatility organic compounds (Bannan et al., 2017). As the degree of functionality increases so does the difficulty in predicting the $P^{sat}$ as more intramolecular forces, steric effects, and shielding effects must be

considered. The majority of GCMs designed for estimating $P^{sat}$ of organic compounds were developed for the chemical industry with a focus on monofunctional compounds with $P^{sat}$ on the order of $10^3 – 10^5$ Pa (Bilde et al., 2015). SOA, in contrast, are typically multifunctional compounds with $P^{sat}$ often many orders of magnitude below $10^{-1}$ Pa (Barley and McFiggans, 2010). GCM development, with a focus on the $P^{sat}$ of SOA has to deal with a lack of robust experimental data and, historically, large differences in measurement data depending on the technique and instrument used to acquire the data.

To address this problem Krieger et al. (Krieger et al., 2018) identified a reference data set for validating $P^{sat}$ measurements using the polyethylene glycol (PEG) series. To improve the performance of GCMs when applied to highly functionalised compounds, more data is required that probes both the effect of relative functional group positioning and the effects of interaction between functional groups on $P^{sat}$, such as in the work by Booth et al. (Booth et al., 2012) and Dang et al. (Dang et al., 2019). In this study the solid state saturation vapour pressure ($P_S^{sat}$) and sub-cooled liquid saturation vapour

pressures ($P_L^{sat}$) of three families of nitroaromatic compounds are determined using KEMS, building on the work done by Dang et al. (Dang et al., 2019) and Bannan et al. (Bannan et al., 2017). These include substituted nitrophenols, substituted nitrobenzoic acids and nitrobenzaldehydes. Nitroaromatics are useful tracers for anthropogenic emissions (Grosjean, 1992), and many nitroaromatic compounds are noted to be highly toxic (Kovacic and Somanathan, 2014). Studies quantifying the overall role of nitrogen containing organics on aerosol formation would also benefit from more refined $P^{sat}$ (Duporté et al.,

2016; Smith et al., 2008). Even if mechanistic models perform poorly predicting aerosol mass due to missing process

phenomena, resolving the partitioning is still important. Several studies have reported the observation of methyl nitrophenols (Chow et al., 2016; Kitanovski et al., 2012; Schummer et al., 2009) and nitrobenzoic acids (van Pinxteren and Herrmann, 2007). Nitrobenzaldehydes can form from the photo-oxidation of toluene in a high $NO_x$ environment (Bouya et al., 2017). Both nitrophenols and nitrobenzoic acids were identified in the review paper by Bilde et al. (Bilde et al., 2015) as

compounds of interest and recommendations for further study. Aldehyde groups tend to have little impact on $P^{sat}$ by themselves but the =O of the aldehyde group can act as a hydrogen bond acceptor.

There is a general lack of literature vapour pressure data for nitroaromatic compounds, and despite recent work on nitrophenols by Bannan et al. (Bannan et al., 2017), there is still a lack of data on such compounds in the literature. This is reflected, in part, in the effectiveness of the GCMs to predict the VP of such compounds.

Here we present $P_S^{sat}$ and $P_L^{sat}$ data for 20 nitroaromatic compounds. The $P_S^{sat}$ data was collected using Knudsen effusion mass spectrometry (KEMS) with a sub-cooled correction performed with thermodynamic data from a differential scanning calorimeter (DSC). The trends in the $P_S^{sat}$ data are considered and chemical explanations are given to explain the observed differences.

As identified by Bilde et al. (Bilde et al., 2015), experimental $P^{sat}$ can differ by several orders of magnitude among

techniques. One way of mitigating this is to collect data for a compound using multiple techniques, whilst running reference compounds to assess consistency among the employed methods. We therefore use supporting data from the electro dynamic balance (EDB) at ETH Zurich for three of the nitroaromatic compounds.

The $P_L^{sat}$ data is then compared with the predicted $P_L^{sat}$ of the GCMs, highlighting where they perform well and where they perform poorly. Finally, these measurements using the new PEG reference standards are compared to past KEMS

measurements using an old reference standard due to differences in experimental $P^{sat}$ between this work and previous KEMS work.

## 2 Experimental

Compound Selection

A total of 10 nitrophenol compounds were selected for this study including 9 monosubstituted, 4 nitrobenzaldehydes

including 1 monosubstituted, and 6 nitrobenzoic acids including 5 monosubstituted. The nitrophenols are shown in Table

1, the nitrobenzaldehydes are shown in Table 2, and the nitrobenzoic acids are shown in Table 3. All compounds selected

for this study were purchased at a purity of 99% and were used without further preparation. All compounds are solid at

room temperature.

### 2.1 Knudsen effusion mass spectrometry system (KEMS)

The KEMS system is the same system that has been used in previous studies (Bannan et al., 2017; Booth et al., 2009,

2010) and a summary of the measurement procedure will be given here. For a more detailed overview see Booth et al.

(Booth et al., 2009). To calibrate the KEMS, a reference compound of known $P^{sat}$ is used. In this study the polyethylene

glycol series (PEG series), PEG-3 ($P_{298} = 6.68 \times 10^{-2}$ Pa) and PEG-4 ($P_{298} = 1.69 \times 10^{-2}$ Pa) (Krieger et al., 2018), were used.

The PEG series is a homologous series that covers 5 orders of magnitude from $10^{-2}$ to $10^{-7}$ Pa and includes PEG-3 through

PEG-8. The $P^{sat}$ of the PEG series were determined using multiple different techniques including multiple electrodynamic

balances (EDBs), a flow tube tandem differential mobility analyser system (FT-TDMA), and a Knudsen effusion mass

spectrometry system (KEMS) (Krieger et al., 2018). By using multiple different techniques it was possible to identify the

lower limits of detection as these were typically where the deviations between measured values occurred. By corroborating

expected trends and absolute values with other methods, it was found that the KEMS was able to determine $P^{sat}$ of PEG-4 to

PEG-7, through good agreement with the other techniques, yet did not capture the expected value of PEG-8. For PEG-8 the

$P^{sat}$ was determined to be 9.2E-08 Pa at 298 K using the EDB and the KEMS system. The KEMS system showed almost no

temperature dependence, which may indicate that the lower limit of detection has been reached at these $P^{sat}$ (Krieger et al.,

2018).

The PEG series has now been employed by new techniques such as, those in Booth et al. (Booth et al., 2017) and Bannan et

al. (Bannan et al., 2019).

The reference compound is placed in a temperature controlled Knudsen cell. The cell has a chamfered orifice through which

the sample effuses creating a molecular beam. The size of the orifice is ≤1/10 the mean free path of the gas molecules in the

cell. This ensures that the particles effusing through the orifice do not disturb the thermodynamic equilibrium of the cell. The


molecular beam is then ionised using a standard 70 eV electron impact ionisation, and analysed using a quadrupole mass

spectrometer.

After correcting for the ionisation cross section (Booth et al., 2009) the signal generated is proportional to the $P^{sat}$. Once the calibration process is completed it is possible to measure a sample of unknown $P^{sat}$. When the sample is changed it is necessary to isolate the sample chamber from the measurement chamber using a gate valve so that the sample chamber can be vented, whilst the ioniser filament and the secondary electron multiplier (SEM) detector can remain on and allow for

direct comparisons with the reference compound. The $P^{sat}$ of the sample can be determined from the intensity of the mass spectrum, if the ionisation cross section at 70 eV, and the temperature at which the mass spectrum was taken are known. The samples of unknown $P^{sat}$ are typically solid so it is the $P_S^{sat}$ that is determined. After the $P_S^{sat}$ (Pa), has been determined for multiple temperatures, the Clausius-Clapeyron equation (Eq. 1) can be used to determine the enthalpy and entropy of sublimation as shown in Booth et al. (Booth et al., 2009).

$$\ln(P^{sat}) = \frac{\Delta H_{sub}}{RT} + \frac{\Delta S_{sub}}{R} \tag{1}$$

where T is the temperature (K), R is the ideal gas constant (J mol$^{-1}$ K$^{-1}$), $\Delta H_{sub}$ is the enthalpy of sublimation (J mol$^{-1}$) and $\Delta S_{sub}$ is the entropy of sublimation (J mol$^{-1}$ K$^{-1}$). $P^{sat}$ was obtained over a range of 30 K in this work starting at 298 K and rising to 328 K. The reported solid state vapour pressures are calculated from a linear fit of ln($P^{sat}$) vs 1/T using the Clausius-Clapeyron equation.

**2.2 Differential scanning calorimetry (DSC)**

According to the reference state used in atmospheric models, and as predicted by GCMs, $P_L^{sat}$ is required. Therefore it is necessary to convert the $P_S^{sat}$ determined by the KEMS system into a $P_L^{sat}$. As with previous KEMS studies (Bannan et al., 2017; Booth et al., 2010) the melting point ($T_m$) and the enthalpy of fusion ($\Delta H_{fus}$) are required for the conversion. These values were measured with a TA Instruments DSC 2500 Differential Scanning Calorimeter (DSC). Within the DSC, heat

flow and temperature were calibrated using an indium reference, and heat capacity using a sapphire reference. A heating rate of 10 K min$^{-1}$ was used. 5-10 mg of sample was measured using a microbalance and then pressed into a hermetically sealed aluminium DSC pan. A purge gas of N$_2$ was used with a flow rate of 30 mL min$^{-1}$. Data processing was performed using



the 'Trios' software supplied with the instrument. $\Delta c_{p,sl}$ was estimated using $\Delta c_{p,sl} = \Delta S_{fus}$ (Grant et al., 1984; Mauger et al., 1972).

### 2.3 Electrodynamic balance (EDB)

The EDB from ETH Zurich has been used to investigate $P^{sat}$ of low volatility compounds in the past (Huisman et al., 2013; Zardini et al., 2006; Zardini and Krieger, 2009) and a brief overview will be given here. For full details see Zardini et al. (Zardini et al., 2006) and Zardini and Krieger (Zardini and Krieger, 2009). The EDB can be applied to both liquid particles and non-spherical solid particles (Bilde et al., 2015). The EDB uses a double ring configuration (Davis et al., 1990) to levitate a charged particle in a cell with a gas flow free from the evaporating species under investigation. There is precise control of both temperature and relative humidity within the cell. Diffusion-controlled evaporation rates of the levitated particle are measured at a fixed temperature and relative humidity by precision sizing using optical resonance spectroscopy in backscattering geometry with a broadband LED source and Mie theory for the analysis (Krieger et al., 2018). $P^{sat}$ is calculated at multiple temperatures and the Clausius-Clapeyron equation can be used to calculate $P^{sat}$ at a given temperature (Eq. 1).

## 3 Theory

### 3.1 Sub-cooled correction

The conversion between $P_S^{sat}$ and $P_L^{sat}$ is done using the Prausnitz equation (Prausnitz et al., 1998) (Eq. 2)

$$\ln\left(\frac{P_L^{sat}}{P_S^{sat}}\right) = \frac{\Delta H_{fus}}{RT_m}\left(\frac{T_m}{T} - 1\right) - \frac{\Delta c_{p,sl}}{R}\left(\frac{T_m}{T} - 1\right) + \frac{\Delta c_{p,sl}}{R}\ln\left(\frac{T_m}{T}\right) \qquad (2)$$

where $P_L^{sat}/P_S^{sat}$ is the ratio between $P_L^{sat}$ and $P_S^{sat}$, $\Delta H_{fus}$ is the enthalpy of fusion (J mol$^{-1}$), $\Delta c_{p,sl}$ is the change in heat capacity between the solid and liquid states (J mol$^{-1}$ K$^{-1}$), T is the temperature (K) and $T_m$ is the melting point (K).

### 3.2 Vapour pressure predictive techniques

The most common $P^{sat}$ prediction techniques are GCMs. Several different GCMs have been developed (Moller et al., 2008; Myrdal and Yalkowsky, 1997; Nannoolal et al., 2008; Pankow and Asher, 2008) with some being more general and others, such as the EVAPORATION method (Compernolle et al., 2011), having been developed with OA as the target compounds.





The Myrdal and Yalkowsky method (Myrdal and Yalkowsky, 1997), the Nannoolal et al. method (Nannoolal et al., 2008), and the Moller et al. method (Moller et al., 2008) are combined methods requiring a boiling point, $T_b$, as an input. If the $T_b$ of a compound is known experimentally it is an advantage, but most atmospherically relevant compounds have an unknown $T_b$ so the $T_b$ that is used as an input is calculated using a GCM. The combined methods use a $T_b$ calculated using a GCM

for many of the same reasons that GCMs are used to calculate $P^{sat}$, i.e. the difficulty in acquiring experimental data for highly reactive compounds or compounds with short lifetimes. The Nannoolal et al. method (Nannoolal et al., 2004), Stein and Brown method (Stein and Brown, 1994), and Joback and Reid method (Joback et al., 1987) are most commonly used. The Joback and Reid method is not considered in this paper due to its known biases (Barley and McFiggans, 2010) and the Stein and Brown method being an improved version of Joback and Reid. The $T_b$ used in the combined methods is, however,

another source of potential error and for methods that extrapolate $P^{sat}$ from $T_b$, the size of this error increases with increasing difference between $T_b$ and the temperature to which it is being extrapolated (O 'meara et al., 2014). EVAPORATION (Compernolle et al., 2011) and SIMPOL (Pankow and Asher, 2008) do not require a boiling point, only requiring a structure and a temperature of interest. The main limitation for many GCMs, aside from the data required to create and refine them, is not accounting for intramolecular interactions, such as hydrogen bonding, or steric effects. The Nannoolal et al.

method (Nannoolal et al., 2008), Moller et al. method (Moller et al., 2008), and EVAPORATION (Compernolle et al., 2011) attempt to address this by having secondary interaction terms. In the Nannoolal et al. method (Nannoolal et al., 2008), there are terms to account for -ortho, -meta, -para isomerism of aromatic compounds, however there are no terms for dealing with tri- or greater substituted aromatics, and in these instances all isomers give the same prediction. A common misuse of GCMs occurs when a GCM is applied to a compound containing functionality not included in the training set, e.g. using

EVAPORATION (Compernolle et al., 2011) with aromatic compounds or using SIMPOL (Pankow and Asher, 2008) with compounds containing halogens. As the GCM does not have the tools to deal with this functionality it will either misattribute a contribution, in the EVAPORATION (2011) example the aromatic structure would be treated as a cyclical aliphatic structure, or simply ignore the functionality, as is the case when SIMPOL (2008) is used for halogen containing compounds. When selecting a GCM to model $P^{sat}$ it is essential to investigate whether the method is applicable to the

compounds of interest. Of the popular $P^{sat}$ GCMs, the Myrdal and Yalkowsky method (Myrdal and Yalkowsky, 1997)



contains only three nitroaromatic compounds, the Nannoolal et al. method (Nannoolal et al., 2008) contains thirteen, the Moller et al. (Moller et al., 2008) contains no more than fourteen, SIMPOL (Pankow and Asher, 2008) contains twenty five, and EVAPORATION (Compernolle et al., 2011) contains zero. The specific nitroaromatics used by the Nannoolal et al. method and the Moller et al. method are not stated (to the author's knowledge) as the data was taken directly from the

Dortmund Data Bank. Despite the SIMPOL (2008) method containing twenty five nitroaromatic compounds, eleven of these are taken from a gas chromatography method using a single data point from a single data set (Schwarzenbach et al., 1988).

## 4 Results

### 4.1 Solid state vapour pressure

$P_S^{sat}$ measured directly by the KEMS are given in Tables 5, 6 and 7 for the nitrophenols, nitrobenzaldehydes and nitrobenzoic acids respectively. Measurements were made at increments of 5 K from 298 to 328 K (with the exception of compounds that melted during the temperature ramp). The Clausius-Clapeyron equation (Eq. 1) was used to calculate the enthalpies and entropies of sublimation. The melting points of compounds studied are given in Tables 8, 9 and 10 for the nitrophenols, nitrobenzaldehydes and nitrobenzoic acids respectively. 2-nitrophenol was measured between 298 K and 318

K, 3-methyl-4-nitrophenol was measured between 298 K and 313K, 4-methyl-2-nitrophenol was measured between 298 K and 303 K, 5-fluoro-2-nitrophenol was measured between 298 K and 308 K, and 2-nitrobenzaldehyde was measured between 298 K and 313 K. Generally speaking, considering the different groups of compounds as a whole, the nitrobenzaldehydes studied exhibit higher $P_S^{sat}$ (order of magnitude) than the nitrophenols and nitrobenzoic acids studied. This is most likely due to the fact that none of the nitrobenzaldehydes studied herein are capable of undergoing hydrogen

bonding (H-bonding) whilst all of the nitrophenols and nitrobenzoic acids, to varying extents, are capable of hydrogen bonding. The nitrophenols and nitrobenzoic acids studied exhibit a range of overlapping $P_S^{sat}$ so nothing can be inferred when considering these two types of compounds together as groups; therefore the differences *within* each of the groups must be considered.

All functional groups around an aromatic ring either withdraw or donate electron density. This is a result of two major

effects, the inductive effect and the resonance effect, or a combination of the two (Ouellette et al., 2015). The inductive

effect is the unequal sharing of the bonding electron through a chain of atoms within a molecule. A methyl group donates

electron density, relative to a hydrogen atom, so is therefore considered an electron donating group, whereas a chloro group

withdraws electron density and is therefore considered an electron withdrawing group. The resonance effect occurs when a

compound can have multiple resonance forms. In a nitro group, as the oxygen atoms are more electronegative than the

nitrogen atom, a pair of electrons from the nitrogen-oxygen double bond can be moved onto the oxygen atom followed by a

pair of electrons being moved out of the ring to form a carbon-nitrogen double bond and leaving the ring with a positive

charge. This leads to the nitro group acting as an electron withdrawing group. In an amino group, on the other hand, the

hydrogens are not more electro negative than the nitrogen; instead the lone pair on the nitrogen can be donated into the ring,

causing the ring to have a negative charge, and the amino group to act as an electron donating group. Examples of the

inductive effect and the resonance effect are given in Fig. 1 (Ouellette et al., 2015).

Some functional groups, such as an aromatic OH group, can both donate and withdraw electron density at the same time. In

phenol the OH group withdraws electron density via the inductive effect, but it also donates electron density via the

resonance effect. This is shown in Fig. 2. As the resonance effect is typically much stronger than the inductive effect, OH

has a net donation of electron density in phenol (see Fig. 2).

The positioning of the functional groups around the aromatic ring determine to what extent the inductive and resonance

effects occur. The changes in electron density due to the inductive effect and the resonance effect also change the partial

charges on the atoms within the aromatic ring. These changes impact the strength of any potential H-bonds that may form.

For instance, in the case of a functionalised phenol, the partial charge of the phenolic carbon is a major factor in the overall

strength of the H-bond (see Fig. 3). The more positive the partial charge of the phenolic carbon the  stronger the H-bond

formed (Remko and Polcin, 1977). In the work by Remko and Polcin (Remko and Polcin, 1977) the effect on the H-bonding

ability of phenol and its ortho, meta and para methoxy substituted derivatives were investigated. Remko and Polcin found

that the ortho and para substituted phenol had weaker intermolecular H-bonds relative to the unsubstituted phenol. The meta

substituted derivative, however, possessed stronger intermolecular hydrogen bonds than the unsubstituted phenol. This trend

is supported by the experimental work by Stymne et al. (Stymne et al., 1973) which also showed the meta substituted

derivative having a higher H-bond energy relative to the unsubstituted phenol and the para isomer having a lower H-bond

energy. The work by Remko and Polcin (Remko and Polcin, 1977) investigated the H-bonding potential to formamide and

the work by Stymne et al. (Stymne et al., 1973) investigated the H-bonding potential to dimethylacetamide. The H-bond

energies and the partial charge of the phenolic OH are shown in Table 4 and the chemical structures of the methoxyphenols

are shown in Fig. 3.

The increase or decrease of the H-bond energy relative to the unsubstituted phenol matches an increase or decrease in the

partial charge of the phenolic carbon. There is a slight discrepancy between 2-methoxyphenol and the 4-methoxyphenol

where 2-methoxyphenol has a higher H-bond energy, but a lower partial charge of the phenolic carbon than 4-

methoxyphenol. This is likely due to 2-methoxyphenol being capable of forming an intramolecular H-bond, which whilst

being weak and the intermolecular H-bond dominating (Remko and Polcin, 1977), will still impact the calculated partial

charge.

Considering first the nitrophenols, Table 5, the highest $P_S^{sat}$ compound is 2-fluoro-4-nitrophenol (2.75E-02 Pa). There are

two potential H-bonding explanations for why this compound has such a high $P_S^{sat}$ relative to the other nitrophenols and

fluoro nitrophenols. First, in this isomer the presence of the F atom on the C adjacent to the OH group gives rise to

intramolecular H-bonding (Fig. 4 left) which reduces the extent of intermolecular interaction possible and increases $P_S^{sat}$.

This effect can clearly be seen from the fact that in 3-fluoro-4-nitrophenol, where the F atom is positioned further away from

the OH group, the $P_S^{sat}$ is significantly lower (4.55E-03) due to the fact that intermolecular H-bonding can occur (Fig. 4

right).

However, in the work by Shugrue et al. (Shugrue et al., 2016) it is stated that neutral organic fluoro and nitro groups form

very weak hydrogen bonds, which whilst they do exist, can be difficult to even detect by many conventional methods. The

second explanation depends on the inductive effect mentioned previously. By using MOPAC2016 (Stewart, 2016), a semi

empirical quantum chemistry program based on the neglect of diatomic differential overlap (NDDO) approximation (Dewar

and Thiel, 1977), the partial charges of the phenolic carbon can be calculated. The partial charge of the phenolic carbon can

be dependent on the orientation of the OH if the molecule doesn't have a plane of symmetry, so in this work the partial

charge used is an average of the two extreme orientations of the OH, as shown in Fig. 5.

The partial charge of the phenolic carbon in 2-fluoro-4-nitrophenol is 0.275 with a $P_S^{sat}$ of 2.75E-02 Pa, whereas for 3-

fluoro-4-nitrophenol it is 0.379 with a $P_S^{sat}$ of 4.55E-03 Pa. The more positive the partial charge of the phenolic carbon the

better it is able to stabilise the increased negative charge which will develop on the O atom as a result of H-bond formation.

As a result stronger intermolecular H-bonds are formed, therefore giving rise to a lower $P_S^{sat}$. Moving the nitro group from

being para to the OH in 3-fluoro-4-nitrophenol to meta to the OH in 5-fluoro-2-nitrophenol further reduces the $P_S^{sat}$ to 4.25E-

03 Pa. This reduction in $P_S^{sat}$ can also be explained via the combination of the inductive effect and the resonance effect as the

partial charge of the phenolic carbon rises from 0.379 to 0.396, again implying stronger intermolecular H-bonds and,

therefore, a lower $P_S^{sat}$.

Similar trends occur in the methyl nitrophenols as in the fluoro nitrophenols with a larger partial charge of the phenolic

carbon corresponding to a lower $P_S^{sat}$. 3-methyl-4-nitrophenol has the most positive partial charge with 0.362 and the lowest

$P_S^{sat}$ of 1.78E-03 Pa, 4-methyl-2-nitrophenol has the next most positive partial charge of 0.343 and the next lowest $P_S^{sat}$ of

3.11E-03, and 4-methyl-3-nitrophenol has the least positive partial charge of 0.249 and the highest $P_S^{sat}$ of 1.08E-02. 3-

methyl-2-nitrophenol does not follow this trend, however, with it having a partial charge of 0.378 and a $P_S^{sat}$ of 9.90E-03. A

possible explanation as to why 3-methyl-2-nitrophenol does not follow this same trend is the positioning of its functional

groups. As shown in Fig. 6 (left), all of the functional groups are clustered together and the proximity of the functional

groups sterically hinders the formation of H-bonds, thus increasing the $P_S^{sat}$. Conversely as shown in Fig. 6 (right) the fact

that the methyl group is further away in 4-methyl-2-nitrophenol leads to less steric hindrance of H-bond formation.

Replacing the methyl group on 4-methyl-2-nitrophenol with an amino group to form 4-amino-2-nitrophenol surprisingly

increases the $P_S^{sat}$ from 3.11E-03 Pa to 3.36E-03 Pa. This is unexpected, as unlike 4-methyl-2-nitrophenol, 4-amino-2-

nitrophenol contains two H-bond donors and so would be expected to have a lower $P_S^{sat}$. The higher $P_S^{sat}$ can be explained

via the combination of the inductive effect and the resonance effect. Whilst the partial charge of the phenolic carbon in 4-

methyl-2-nitrophenol is 0.343, the partial charge of the phenolic carbon in 4-amino-2-nitrophenol is only 0.264 and the

partial charge of the carbon bonded to the amine group is only 0.211. So whilst 4-amino-2-nitrophenol is capable of forming two intermolecular H-bonds compared to 4-methyl-2-nitrophenol's one, they will be much weaker.

Replacing the methyl group on 4-methyl-3-nitrophenol with a chloro group to form 4-chloro-3-nitrophenol reduces the $P_S^{sat}$ from 1.08E-02 Pa to 2.26E-03 Pa. This reduction in $P_S^{sat}$ can be explained by the increase in partial charge of the phenolic carbon from 0.249 to 0.266, as well as a 13% increase in molecular weight.

Replacing the F atom in 3-fluoro-4-nitrophenol with a methyl group to form 3-methyl-4-nitrophenol further reduces the $P_S^{sat}$ (1.78E-03) although exactly why is unclear. The methyl group cannot engage in intermolecular H-bonding, it will sterically hinder any H-bonding that the $NO_2$ group undergoes and it reduces the net dipole moment of the molecule (from 6.36 D to 5.41 D) (Stewart, 2016) which would reduce the extent of dipole-dipole type interactions between the molecules. The net dipole moments were calculated using MOPAC2016 (Stewart, 2016), and similarly to the partial charges, are an average taken from the extreme orientations of the OH group, aldehyde groups, or carboxylic acid group. It is possible that the crystallographic packing density of 3-methyl-4-nitrophenol is higher although no data is available to support this.

Removing the methyl groups from both 3-methyl-2-nitrophenol and 4-methyl-2-nitrophenol to give 2-nitrophenol causes the $P_S^{sat}$ to drop from 9.90E-03 Pa and 3.11E-03 Pa, respectively, to 8.94E-04 Pa. This reduction in $P_S^{sat}$ matches an increase in the positive partial charge of the phenolic carbon, from 0.378 and 0.343 to 0.383, implying an increase in the strength of the intermolecular H-bonds and therefore a reduction in $P_S^{sat}$.

Now considering the nitrobenzaldehydes (Table 6) the highest $P_S^{sat}$ compound is 2-nitrobenzaldehyde (3.32E-01). Comparing this to 2-nitrophenol (8.94E-04) shows how significant the ability to form H-bonds is to the $P_S^{sat}$ of a compound, with replacing a hydroxyl group (capable of H-bonding) with an aldehyde group (incapable of H-bonding) raising the $P_S^{sat}$ of the compound by more than two orders of magnitude. The decrease in $P_S^{sat}$ observed by moving the nitro group from being ortho to the aldehyde group in 2-nitrobenzaldehyde, to being meta in 3-nitrobenzaldehyde (1.21E-01) and para in 4-nitrobenzaldehyde (3.40E-02) can be explained using the different crystallographic packing densities of the three isomers. Crystallographic packing density is a measure of how densely packed the molecules of a given compound are when they crystallise, the more closely packed molecules are the greater the overall extent of interaction between them and the lower the $P_S^{sat}$. The order of the $P_S^{sat}$ observed here for the three isomers of nitrobenzaldehyde matches that of their crystallographic



packing densities (Coppens and Schmidt, 1964; Engwerda et al., 2018; King Jnr and Bryant Jnr, 1996), with the lowest $P_S^{sat}$ correlating with the highest packing density and vice versa.

The addition of a Cl atom to 3-nitrobenzaldehyde is also observed to decrease the compounds $P_S^{sat}$. This can be simply

rationalised due to the greater than 25% increase this causes to the molecular weight. The higher a compounds molecular weight the greater the overall extent of interaction between its molecules and the lower its $P_S^{sat}$.

The trend of the nitrobenzaldehyde $P_S^{sat}$ matches the measured melting point trend shown in Table 9, where 2-nitrobenzaldehyde has the highest $P_S^{sat}$ (3.54E-01 Pa) and the lowest melting point (44.51 ∘C) and 4-nitrobenzaldehyde has the lowest $P_S^{sat}$ (3.40E-02 Pa) and the highest melting point (107.25 ∘C).

Finally considering the nitrobenzoic acids (Table 7), the highest $P_S^{sat}$ compound is 4-methyl-3-nitrobenzoic acid (4.67E-03). Its isomer 3-methyl-4-nitrobenzoic acid possesses a slightly lower $P_S^{sat}$ (3.97E-03) which could be attributed to the slight increase in the partial charge of the carbon within the carboxylic acid group (from 0.628 to 0.644) although the difference is not significant. 4-methyl-3-nitrobenzoic acid and 3-methyl-4-nitrobenzoic acid both exhibit lower $P_S^{sat}$s than their corresponding nitrophenols (4-methyl-3-nitrophenol and 3-methyl-4-nitrophenol respectively) which demonstrates the

increased suppressive effect on $P_S^{sat}$ that carboxylic acid groups have compared to hydroxyl groups. This is due to the fact that carboxylic acid groups allow for a molecule to H-bond to three neighbouring molecules (Fig. 7 left), whilst a hydroxyl group allows for only two H-bonds (Fig. 7 right), and this increased extent of intermolecular interaction leads to a reduction in $P_S^{sat}$.

Removing the methyl group from 4-methyl-3-nitrobenzoic acid to give 3-nitrobenzoic acid (1.10E-03) reduces the observed

$P_S^{sat}$ most likely due to the reduction in steric hindrance around the nitro group which would allow for more effective H-bonding. In addition 3-nitrobenzoic acid possesses a lower $P_S^{sat}$ than the corresponding 3-nitrobenzaldehyde due to its ability to form H-bonds. Adding a hydroxyl group or a Cl atom to 3-nitrobenzoic acid to give 2-hydroxy-5-nitrobenzoic acid (1.79E-03) or 2-chloro-3-nitrobenzoic acid (1.97E-03) respectively increases the observed $P_S^{sat}$ as the addition of the extra functional group leads to increased intramolecular H-bonding occurring. Additionally, comparing 2-hydroxy-5-nitrobenzoic

acid with 2-fluoro-4-nitrophenol demonstrates how the increased ability of carboxylic acid to partake in H-bonding compared to a F atom leads to a suppression of $P_S^{sat}$. 5-Chloro-2-nitrobenzoic acid has a higher $P_S^{sat}$ (2.98E-03 Pa) than 2-

chloro-3-nitrobenzoic acid (1.97E-03 Pa), its structural isomer. The increase in $P_S^{sat}$ can be attributed to the increase partial charge of the carbon within the carboxylic acid group (0.627 increasing to 0.640).

In summary the ability to form H-bonds appears to be the most significant factor affecting the $P_S^{sat}$ of a compound, where
molecules that are able to form these strong intermolecular interactions generally always exhibit lower $P_S^{sat}$ than those that cannot. Additionally different functional groups are able to form different numbers of H-bonds; with those that are able to form more H-bonds generally supressing $P_S^{sat}$ to a greater extent than those that form less. The relative positioning of those functional groups responsible for the H-bonding is also important as when positioned too close together intramolecular H-bonding can occur, which competes with intermolecular H-bonding and generally raises $P_S^{sat}$. The positioning of non H-
bonding functional groups within the molecule can also have an impact upon the extent of H-bonding, with bulky substituents positioned close to H-bonding groups causing steric hindrance which reduces the extent of H-bonding and generally raises $P_S^{sat}$. The positioning of all the functional groups around the aromatic ring effect the partial charges of the atoms, via a combination of the inductive effect and the resonance effect. The inductive effect and the partial charges appear to be most important when comparing isomers, and less important when one functional group has been swapped for another.
In addition higher molecular dipole moments, greater molecular weight, and increased crystallographic packing density also negatively correlate with $P_S^{sat}$ as they all lead to increased overall intermolecular interactions. However in many cases these different factors compete with each other, making it difficult to predict the expected $P_S^{sat}$ and currently it is not possible to determine which factor will dominate in any given case.

## 4.2 Sub-cooled liquid vapour pressure

The $P_L^{sat}$ were obtained from the $P_S^{sat}$ using thermochemical data obtained through use of a DSC and Eq. 2. The results are detailed in Tables 8, 9 and 10 for the nitrophenols, nitrobenzaldehydes and nitrobenzoic acids respectively.

Comparing the $P_L^{sat}$ of the nitrophenols with the solid state values there are a few changes in the overall ordering but they mostly have little effect upon the preceding discussion. A few previously significant increases/decreases in $P^{sat}$ become insignificant and a few that were insignificant are now significant. One point of note however, is that 3-methyl-4-nitrophenol
(5.86E-02) now exhibits a higher $P^{sat}$ than 3-fluoro-4-nitrophenol (3.32E-02). This trend is what would be expected based

on the reduction in steric hindrance, increased potential for H-bonding and increase in molecular dipole moment that the F atom provides in comparison to the methyl group.

For the nitrobenzaldehydes one change in the overall ordering of the $P^{sat}$s is observed after converting to $P_L^{sat}$ but this has no effect on the preceding discussion.

Finally for the nitrobenzoic acids whilst some previously insignificant differences in $P_S^{sat}$ have now become significant, the only change that impacts upon the discussion is that the $P^{sat}$ of 3-methyl-4-nitrobenzoic acid (3.04E-01) is now higher than that of 4-methyl-3-nitrobenzoic acid (5.76E-02). This change could be explained as a result of the higher molecular dipole moment of 4-methyl-3-nitrobenzoic acid (4.306 D vs 3.555 D) (Stewart, 2016) playing a more important role in the subcooled liquid state than in the solid state.

**4.3 Comparison with estimations from GCMs**

In Fig. 8 the experimentally determined $P_L^{sat}$ of the nitroaromatics are compared to the predicted values of several GCMs. These GCMs are SIMPOL (Pankow and Asher, 2008), EVAPORATION (Compernolle et al., 2011), the Nannoolal et al. method (Nannoolal et al., 2008), and the Myrdal and Yalkowsky method (Myrdal and Yalkowsky, 1997). The Nannoolal et al. method (Nannoolal et al., 2008) and the Myrdal and Yalkowsky method (Myrdal and Yalkowsky, 1997) are both 385 combined methods which require a boiling point to function. As for many compounds where the experimental boiling point is unknown boiling point group contribution methods are required. The Nannoolal et al. method (Nannoolal et al., 2004) and the Stein and Brown method (Stein and Brown, 1994) are used.

The Myrdal and Yalkowsky method (Myrdal and Yalkowsky, 1997) shows poor agreement with the experimental data for almost all compounds, but is not particularly surprising given that it only contains 3 nitroaromatic compounds in this 390 method's fitting data set, with none of these compounds containing both a nitro group and another oxygen containing group. The Myrdal and Yalkowsky method (Myrdal and Yalkowsky, 1997) is the oldest method examined in this study, and much of the atmospherically relevant $P^{sat}$ data has been collected after the end of the development of this model. The Myrdal and Yalkowsky method's (Myrdal and Yalkowsky, 1997) reliance on a predicted boiling point may also be a major source of error in the $P^{sat}$ predictions of the nitroaromatics.



On average the SIMPOL method (Pankow and Asher, 2008) predicts values closest to the experimental data, on average predicting $P_L^{sat}$ 1.3 orders of magnitude higher than the experimental values, despite absolute differences of up to 4.4 orders of magnitude.

EVAPORATION (Compernolle et al., 2011) has the worst agreement with the experimental data, on average predicting $P_L^{sat}$ 3.9 orders of magnitude higher than the experimental values and absolute differences of up to 7.0 orders of magnitude. This

outcome is not unexpected because, whilst EVAPORATION (Compernolle et al., 2011) was designed with SOAs in mind, it does not contain any aromatic parameters and is therefore unsuitable for any aromatic compounds. It has been used to demonstrate the effects of using GCMs that do not contain the functionality of the compounds of interest and the large errors in estimation that this can cause.

The Nannoolal et al. method (Nannoolal et al., 2004) is persistently worse than the Stein and Brown method (Stein and

Brown, 1994) for the nitroaromatic compounds involved in this study. When discussing the Nannoolal et al. method (Nannoolal et al., 2008) and the Myrdal and Yalkowsky method (Myrdal and Yalkowsky, 1997) from this point onwards it is used with the Stein and Brown method (Stein and Brown, 1994) unless stated otherwise.

The Nannoolal et al. method (Nannoolal et al., 2008) has slightly better agreement with the experimental data when compared to the Myrdal and Yalkowsky method (Myrdal and Yalkowsky, 1997) on average predicting $P_L^{sat}$ 2.52 orders of

magnitude higher than the experimental values, whereas the Myrdal and Yalkowsky method (Myrdal and Yalkowsky, 1997) on average predicts $P_L^{sat}$ 2.65 orders of magnitude higher than the experimental values. The Nannoolal et al. method (Nannoolal et al., 2008), unlike the others, contains parameters for ortho, meta, para isomerism and even demonstrates the same trend as the experimental data for 2-nitrobenzaldehyde, 3-nitrobenzaldehyde and 4-nitrobenzaldehyde, although 3 orders of magnitude higher. Despite the ortho, meta, para parameters, as soon as a third functional group is present around

the aromatic ring the Nannoolal et al. method (Nannoolal et al., 2008) no longer accounts for relative positioning of the functional groups.

Figure 8a shows the comparison between the experimental and predicted $P_L^{sat}$ for the nitrophenols. Both SIMPOL (Pankow and Asher, 2008) and the Nannoolal et al. method (Nannoolal et al., 2008) contain nitrophenol data from Schwarzenbach et al. (Schwarzenbach et al., 1988). This data of Schwarzenbach et al. (Schwarzenbach et al., 1988), however, is questionable

in reliability due to being taken from a single data point from a single data set. The values given are also 3-4 orders of

magnitude greater than those measured in this work as well as those measured by Bannan et al. (Bannan et al., 2017) and

those measured by Dang et al. (Dang et al., 2019) The use of the Schwarzenbach et al. (Schwarzenbach et al., 1988)

nitrophenol $P^{sat}$ data, which makes up 11 of the 12 nitrophenol data points within the fitting data set of the SIMPOL method

(Pankow and Asher, 2008), is a likely cause of the SIMPOL method (Pankow and Asher, 2008) overestimating the $P^{sat}$ of

nitrophenols by 3 to 4 orders of magnitude. The one nitrophenol used in the SIMPOL method(Pankow and Asher, 2008) not

from Schwarzenbach et al. (Schwarzenbach et al., 1988), 3-nitrophenol from Ribeiro da Silva et al. (Ribeiro da Silva et al.,

1992), has a much lower $P^{sat}$ than those of Schwarzenbach et al. and is only one order of magnitude higher than that from

Bannan et al. (Bannan et al., 2017). Additionally, Whilst the Nannoolal et al. (Nannoolal et al., 2008) method performs

slightly better than the Myrdal and Yalkowsky method (Myrdal and Yalkowsky, 1997) overall for this study, when taking

the nitrophenol data in isolation this performance is flipped with the Myrdal and Yalkowsky method (Myrdal and

Yalkowsky, 1997) showing better performance (overestimating on average by 3.4 to 3.5 orders of magnitude).

Figure 8b shows the comparison between the experimental and predicted $P_L^{sat}$ for the nitrobenzaldehydes. There are no

nitrobenzaldehydes present in any fitting data set of the GCMs considered in this study. Despite this, whilst not capturing the

effects of ortho, meta, para isomerism, SIMPOL (Pankow and Asher, 2008) predicts the $P^{sat}$ of the nitrobenzaldehydes to,

on average, 0.29 orders of magnitude. As polar groups such as aldehydes have been shown to have little impact on volatility

in the pure component, and by extension $P^{sat}$ (Bilde et al., 2015), this implies that SIMPOL (Pankow and Asher, 2008)

captures the contribution of the nitro group very well. Similar to the nitrophenols the performance of the Nannoolal et al.

method (Nannoolal et al., 2008) and the Myrdal and Yalkowsky method (Myrdal and Yalkowsky, 1997) has switched for the

nitrobenzaldehydes compared to the entire data set. The Myrdal and Yalkowsky method (Myrdal and Yalkowsky, 1997)

overestimates by 2.4 orders of magnitude compared to the Nannoolal et al. method (Nannoolal et al., 2008) which

overestimates by 2.5 orders of magnitude.

Figure 8c shows the comparison between the experimental and predicted $P_L^{sat}$ for the nitrobenzoic acids. SIMPOL (Pankow

and Asher, 2008) contains, though in limited amounts, nitrobenzoic acid data in its fitting parameters. Although there are no

lists of the data used to form the Nannoolal et al. method (Nannoolal et al., 2008) available (to the authors knowledge), it is



stated that the values come from the Dortmund Data Bank and from searches on this database there is nitrobenzoic acid $P^{sat}$ data available. Having even this limited data available for the nitrobenzoic acids allows for SIMPOL (Pankow and Asher, 2008) to predict the $P_L^{sat}$s of 5-chloro-2-nitrobenzoic acid, 3-nitrobenzoic acid, 2-chloro-3-nitrobenzoic acid and 2-hydroxy-5-nitrobenzoic acid to within one order of magnitude of the experimental values. On average the SIMPOL (Pankow and Asher, 2008) method underestimates $P_L^{sat}$ by 0.8 orders of magnitude. The nitrobenzoic acids that had large discrepancies

with SIMPOL (Pankow and Asher, 2008), 4-methyl-3-nitrobenzoic acid and 3-methyl-4-nitrobenzoic acid, as well as 2-hydroxy-5-nitrobenzoic acid agreed to within one order of magnitude of the Nannoolal et al. method (Nannoolal et al., 2008). On average the Nannoolal et al. method (Nannoolal et al., 2008) overestimates $P_L^{sat}$ by 0.9 orders of magnitude.

Overall SIMPOL (Pankow and Asher, 2008) performs relatively well for the nitrobenzaldehydes and the nitrobenzoic acids, and the Nannoolal et al. method (Nannoolal et al., 2008) performs moderately well for the nitrobenzoic acids when

compared to the experimental values found in this study. All of the methods perform poorly when compared to the experimental nitrophenol values. These observations are not particularly surprising when taking into account how the methods were fitted and what data is present in the fitting set.

### 4.4 Comparison with existing experimental data

For the compounds in this study that had previous literature data there are differences from the values determined

experimentally in this work. The differences between the values from this work and those of Dang et al. (Dang et al., 2019) are discussed in sect. 4.5 but can be attributed to the use of a different reference compound.

For the nitrophenols, shown in Fig. 8a, the differences between the experimental values and the literature values from Schwarzenbach et al. (Schwarzenbach et al., 1988) range from 3 to 4 orders of magnitude. The relationship between the $P_L^{sat}$ and temperature from Schwarzenbach et al. (Schwarzenbach et al., 1988) was derived from gas chromatographic (GC)

retention data. This GC method requires a reference compound of known $P^{sat}$, and for the reference compound and the compound of interest to have very similar interactions with the stationary phase of the GC. Schwarzenbach et al. (Schwarzenbach et al., 1988) used 2-nitrophenol as the reference compound for all of the other nitrophenol data they collected. In this work the $P_L^{sat}$ at 298 K was 1.38E-03 Pa whereas Schwarzenbach et al. (Schwarzenbach et al., 1988) reported it as 2.69E+01 Pa. As the difference between the $P^{sat}$ of 2-nitrophenol in this work and  Schwarzenbach et al.

(Schwarzenbach et al., 1988) differs by approximately 4 orders of magnitude this could explain why the other nitrophenol

measurements also differ by 3-4 orders of magnitude.

For the nitrobenzaldehydes, shown in Fig. 8b, the literature data from Perry et al. (Perry et al., 1984) and the experimental

data from this work agree within one order of magnitude with 2-nitrobenzaldehyde especially agreeing very closely

(2.39E+00 Pa vs 2.15E+00 Pa).

The nitrobenzoic acids are shown in Fig. 8c. The value for 3-nitrobenzoic acid from this work is 1.90E-03 Pa compared to

5.05E-03 from Ribeiro da Silva et al. (Ribeiro Da Silva et al., 1999) Whilst not matching perfectly, the $P^{sat}$ of 3-nitrobenzoic

acid is on this order of magnitude. The disagreements between the values of this work and the values from Monte et al.

(Monte et al., 2001) for 4-methyl-3-nitrobenzoic acid and 3-methyl-4-nitrobenzoic acid are quite large. 4-methyl-3-

nitrobenzoic acid differs by over one order of magnitude and 3-methyl-4-nitrobenzoic acid is closer to two orders of

magnitude. Monte et al. (Monte et al., 2001) where collected using a Knudsen mass loss method. Knudsen mass loss is

similar to KEMS in that it also utilises a Knudsen cell which effuses the compound of interest. However for an amount of

mass to be lost such that it can be detected the experiments need to be performed at higher temperatures than the KEMS.

This means that the data must be extrapolated further to reach ambient temperatures. This is a potential source of error and

could explain the difference. Measurement by a third or even fourth technique would be required to confirm this.

**4.5 Sensitivity of vapour pressure measurement techniques to reference standards**

The recently published paper by Dang et al. (Dang et al., 2019) measured the $P^{sat}$ of several of the same compounds that

are studied in this paper using the same KEMS system, however in this study the newly defined best practice reference

sample was used (Krieger et al., 2018), whereas Dang et al. (Dang et al., 2019) used malonic acid. These compounds were 4-

methyl-3-nitrophenol, 3-methyl-4-nitrophenol and 4-methyl-2-nitrophenol. The difference in reference compound led to a

discrepancy in the experimental $P^{sat}$ (shown in Table 11). Due to these differences additional measurements were made

using malonic acid as the reference material. Additionally, supporting measurements for the compounds were performed

using the EDB from ETH Zurich in order to rule out instrumental problem with the KEMS. As single particles injected from

a dilute solution may either stay in a supersaturated, liquid state or crystallize, it is important to identify its physical state.





For 4-methyl-3-nitrophenol a 3 % solution dissolved in isopropanol was injected into the EDB. After the injection and fast

evaporation of the isopropanol, all particles were non-spherical, but with only small deviations from a sphere, meaning that it

was unclear whether the phase was amorphous or crystalline. To determine the phase of this first experiment, a second

experiment was performed, where a solid particle was injected directly into the EDB. Mass loss with time was measured by

following the DC voltage necessary to compensate the gravitational force acting on the particle to keep the particle

levitating. When comparing the $P^{sat}$ from both of these experiments it is clear that the initial measurement of 4-methyl-3-

nitrophenol was in the crystalline phase.

3-methyl-4-nitrophenol was only injected as a solution but the particle crystallized and was clearly in the solid state.

4-methyl-2-nitrophenol was injected as both a 3 % and 10 % solution. Despite being able to trap a particle, the particle

would completely evaporate within about 30 seconds. This evaporation time scale is too small to allow the EDB to collect

any quantitative data. Using the equation for large particles neglecting evaporative cooling (Hinds, 1999) (Eq. 3) it is

possible to estimate $P^{sat}_L$

$$t = \frac{R\rho \cdot d_p{}^2}{8DM\frac{P^{sat}}{T}} \tag{3}$$

where t is the time that the particle was trapped within the cell of the EDB, R is the ideal gas constant, $\rho$ is the density of the

particle, $d_p$ is the diameter of the particle, D is the diffusion coefficient, M is the molecular mass, T is the temperature, and

$P^{sat}$ is the saturation vapour pressure. Eq. 3 gives approximately 4.3E-03 Pa for $P^{sat}_L$ at 290 K.

Comparisons between $P^{sat}$ at 298 K from the KEMS using a PEG reference, the KEMS using a malonic acid reference,

Dang et al. (Dang et al., 2019) and the EDB are shown in Table 11. Following this in Table 12 $P^{sat}_L$, extrapolated down to

290 K, from KEMS using a PEG reference and the KEMS using a malonic acid reference are compared to the estimated $P^{sat}_L$

based on the findings from the EDB using Eq. 3.

Whilst the absolute values of the nitrophenols shown in Table 11 changed, the $P^{sat}$ trends did not. The values from Dang et

al. (Dang et al., 2019) are between 4.39 and 7.81 times lower than those in this work using the PEGs as the reference

compound, which is now deemed as best practice in the community. To ensure that the difference in reference compound

was the cause of the difference in $P^{sat}$ 4-methyl-2-nitrophenol, 4-methyl-3-nitrophenol and 3-methyl-4-nitrophenol were

also measured using malonic acid as a reference again. The differences between the $P^{sat}$ determined by Dang et al. (Dang et

al., 2019) and those in this work using malonic acid as a reference compound were between 2 % and 27 %, well within the

quoted 40 % error margin of the KEMS, (Booth et al., 2009) therefore showing that the instrument is behaving reproducibly but with now improved reference standards being used, as is discussed below.

Starting with 4-methyl-3-nitrophenol the EDB has much better agreement with the KEMS when the PEGs are used as the reference compound than when malonic acid is used as the reference compound. When the quoted errors of both the EDB (shown in Table 11) and the KEMS ($\pm 40\%$ for $P_S^{sat}$ and $\pm 75\%$ for $P_L^{sat}$ (Booth et al., 2009)) are taken into account the lower

limit of the EDB (1.57E-02 Pa) and the upper limit of the KEMS using the PEG references (1.51E-02 Pa) almost overlap whereas the EDB data is almost 1 order of magnitude larger than the KEMS when the malonic acid reference is used (shown in Fig. 9).

For 3-methyl-4-nitrophenol a comparison can be made for both $P_S^{sat}$ and $P_L^{sat}$. Looking first at the $P_S^{sat}$ the EDB appears to be somewhere in between the KEMS depending on what the KEMS is using as a reference, with its absolute value being closer

to that of the Malonic acid reference. However when the quoted errors are taken into account (shown in Table 11) the EDB actually has better agreement with the KEMS when the PEG references are used. This can be seen more clearly in Fig. 9. For $P_L^{sat}$ the EDB and the KEMS when using the PEG references appears to agree very well with a large overlap when the quoted errors are taken into account. This can also be seen in Fig. 9.

The confidence with which the comparison between the EDB and the KEMS can be made for 4-methyl-2-nitrophenol is

lower than with the other compounds looked at due to how quickly 4-methyl-2-nitrophenol evaporated in the EDB. To make this comparison the $P_L^{sat}$ from the KEMS measurements has been extrapolated down to 290 K to match that of the EDB estimation. The predicted EDB value (shown in Table 12) is higher than the KEMS for both references but has a very large error margin (approximately a factor of 5). When this error is considered the KEMS using the PEG reference is within this range, whereas there is close to an order of magnitude difference between the lower limit of this estimate and the upper limit

of the KEMS when malonic acid is used as the reference.

In all cases the EDB showed better agreement with the KEMS using the PEGs as the reference material compared to when malonic acid was used as the reference material. For 4-methyl-3-nitrophenol the agreement was very close between the EDB and the KEMS using the PEGs as the reference compounds, and for 3-methyl-4-nitrophenol the measurements for the EDB and the KEMS agreed with each other within the quoted errors. For 4-methyl-2-nitrophenol the KEMS with PEG as a

reference also showed the best agreement with the EDB, but as this was an estimate with a large error range this comparison

is the least certain.

## 5 Conclusions

Experimental values for the $P_S^{sat}$ and $P_L^{sat}$ have been obtained using KEMS and DSC for nitrophenols, nitrobenzaldehydes,

and nitrobenzoic acids.

The predictive models have been shown to overestimate $P_L^{sat}$ in almost every instance by several orders of magnitude. As the

$P^{sat}$ from these predictive techniques are often used in mechanistic partitioning models (Lee-Taylor et al., 2011; Shiraiwa et

al., 2013), the overestimation of the $P^{sat}$ can lead to an overestimation of the fraction in gaseous state. The experimental

values from this study can be used in conjunction with other measurements to improve the accuracy of GCMs, and give an

insight into the impact of functional group positioning which is missing, or only available in a limited capacity, for the

currently available GCMs.

The differences in trends of the experimental $P^{sat}$ have been explained chemically, with the potential and strength of H-

bonding appearing to be the most significant factor, where present, in determining the $P^{sat}$. With the stronger the hydrogen

bond and the increasing number of possible hydrogen bonds decreasing the $P^{sat}$. Whilst H-bonding is typically the most

important factor, it isn't the only factor. Steric effects by functional groups can also have significant effects on the $P^{sat}$, and

in systems without the potential to form H-bonds the dipole moment of a compound can become important. In the solid state

crystallographic packing density can also be an important factor. To further investigate the impacts of H-bonding, inductive

and resonance effects, and steric effects on $P^{sat}$ more compounds need to be investigated, with select compounds being

chosen to probe these effects.

The predictive models consistently overestimate the $P_L^{sat}$s by up to 6 orders of magnitude with the nitrophenols performing

especially poorly. This demonstrates a need for more experimental data to be used in the fitting data sets of the GCMs to

reduce the errors and give more accurate results for nitroaromatic compounds.

Deviations between the measurements in Dang et al. (Dang et al., 2019) and this work can be explained by the difference of the reference material used which demonstrates the necessity of a consistent, widely used reference compound. The PEG series, looked at by Krieger et al. (Krieger et al., 2018), is currently the preferred reference/calibration series.

Comparisons between the KEMS and the EDB from ETH were made for several nitrophenols. The EDB showed close agreement with the KEMS when the PEG series was used as the reference compounds.

Compounds such as the nitrobenzaldehydes, which are capable of being H-bond acceptors but not H-bond donors, are likely to deviate negatively from Raoult's law in mixtures with compounds that can act as H-bond donors, due to the adhesive forces present. This could call into question the validity of pure component vapour pressure measurements for looking at

atmospheric systems due to the atmosphere not being made up of the pure component. This would be an interesting avenue of research and the natural progression from pure component measurements to investigate their usefulness.

### Data Availability

All data in this paper is available from http://doi.org/10.5281/zenodo.3613581 (Shelley et al., 2020b).

### Supplementary Material

The supplementary material is available from https://doi.org/10.5281/zenodo.3625641 (Shelley et al., 2020a)

### Author Contributions

Petroc D. Shelley carried out the experiments on the KEMS and DSC. Ulrich K. Krieger carried out the experiments on the EDB. Formal analysis of the data was carried out by Petroc D. Shelley, Stephen D. Worrall and Ulrich K. Krieger. Project supervision was undertaken by David Topping, M. Rami Alfarra and Thomas J. Bannan. KEMS training was performed by

Thomas J. Bannan. Access to and training on the DSC was undertaken by Arthur Garforth. Verification on the reliability of the KEMS was carried out by Ulrich K. Krieger, with the EDB measurements being used to validate the KEMS measurements. The original draft manuscript was written by Petroc D. Shelley, Stephen D. Worrall and Carl J. Percival. Internal review and editing was performed by Thomas J. Bannan, David Topping, M. Rami Alfarra, Stephen D. Worrall and Ulrich K. Krieger.



## Competing Interests

The Authors declare that they have no conflict of interest.

## Acknowledgements

The work contained in this paper contains work conducted during a PhD study supported by the Natural Environment Research Council (NERC) EAO Doctoral Training Partnership and is fully-funded by NERC whose support is gratefully acknowledged.

Grant ref no is NE/L002469/1

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





Inductive effect - Electron withdrawing (right) and electron donating (left)

Resonance effect - Electron withdrawing (left) and electron donating (right)

**Figure 1: The inductive effect and the resonance effect**

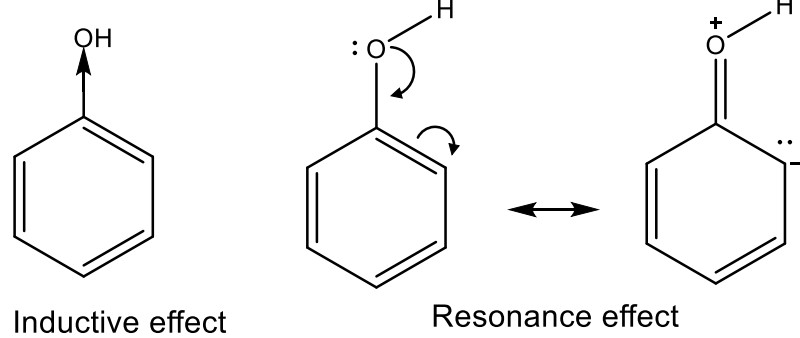

Inductive effect

Resonance effect

**Figure 2: Phenol can withdraw electron density via the inductive effect (left) and donate electron density via the resonance effect**

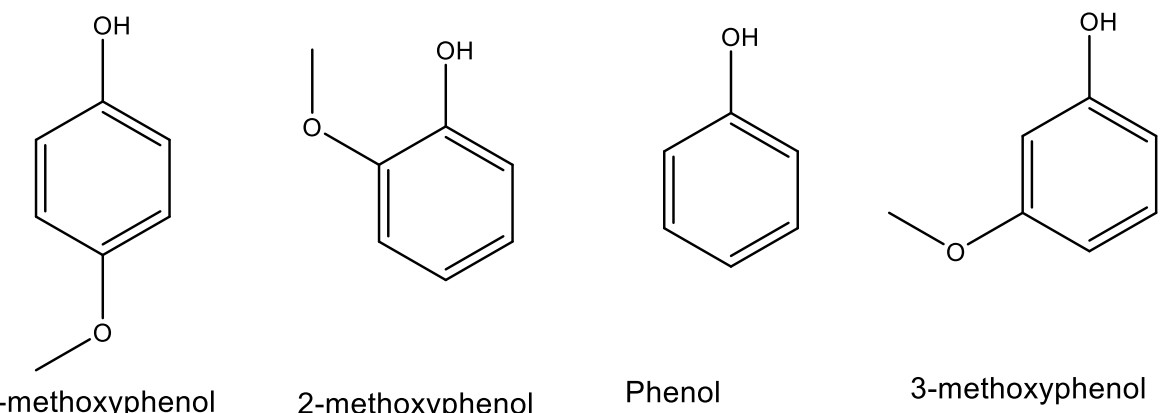

4-methoxyphenol    2-methoxyphenol    Phenol    3-methoxyphenol

**Figure 3: structures of phenol and the methoxyphenol isomers**


**Figure 4: Intramolecular hydrogen bonding in 2-fluoro-4-nitrophenol (left) in comparison to intermolecular hydrogen bonding in**

**3-fluoro-4-nitrophenol**

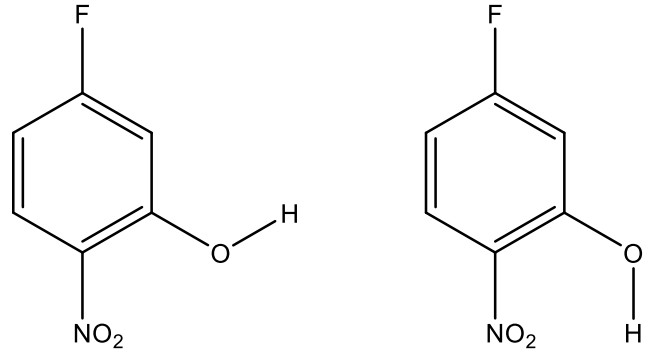

Partial charge of
phenolic carbon: 0.375

Partial charge of
phenolic carbon: 0.417

**Figure 5: The orientation of the OH group can impact the partial charge of the phenolic carbon**





**Figure 6: Diagram emphasising how the proximity of the bulky methyl group sterically hinders intermolecular interactions with the nitro group in 3-methyl-2-nitrophenol (left) but not in 4-methyl-2-nitrophenol (right).**

**Figure 7: Diagram demonstrating how the a carboxylic acid functionality allows a molecule to hydrogen bond to three other**
**molecules in 4-methyl-3-nitrobenzoic acid (left) whilst a hydroxyl group only allows for hydrogen bonding to two other molecules in 4-methyl-3-nitrophenol (right).**



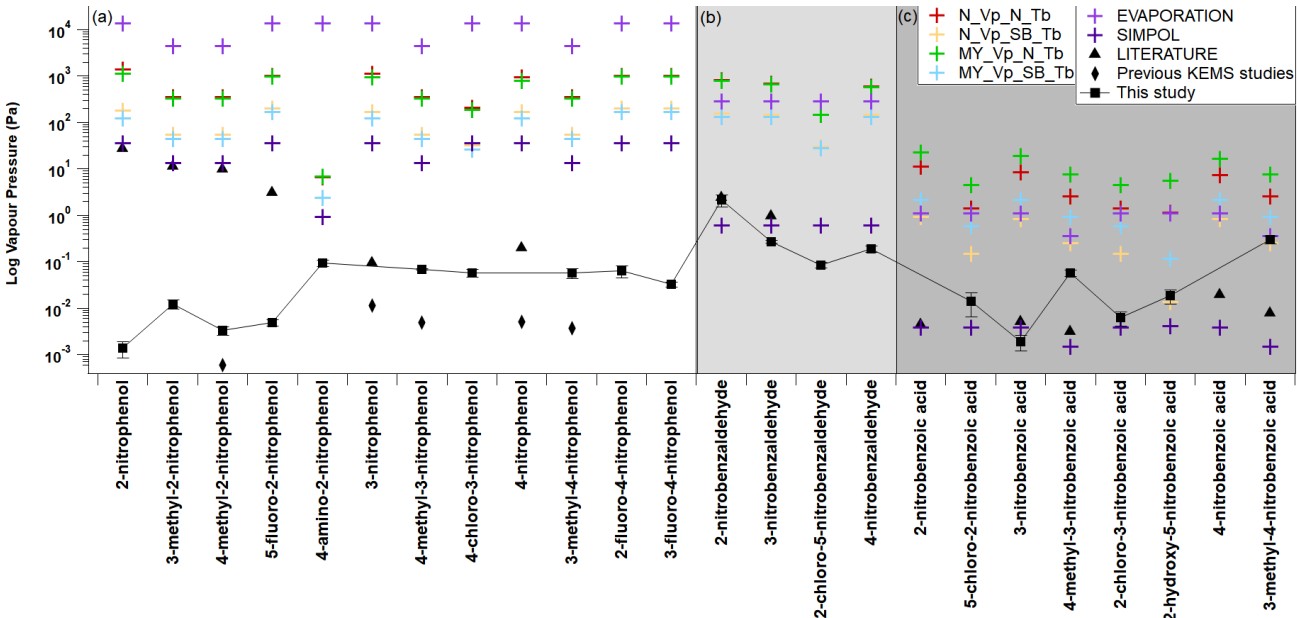

**Figure 8: Comparison of estimated and measured sub-cooled saturation vapour pressures. N_Vp (Nannoolal vapour pressure), MY_Vp (Myrdal and Yalkowsky vapour pressure), EVAPORATION (EVAPORATION vapour pressure), SIMPOL (SIMPOL vapour pressure), N_Tb (Nannoolal boiling point), SB_Tb (Stein and Brown boiling point), LITERATURE - black triangle (2-nitrophenol, 3-methyl-2-nitrophenol, 4-methyl-2-nitrophenol, 5-fluoro-2-nitrophenol, 4-nitrophenol from** (Schwarzenbach et al., 1988)**, 3-nitrophenol from** (Ribeiro da Silva et al., 1992) **2-nitrobenzaldehyde, 3-nitrobenzaldehyde from** (Perry et al., 1984)**, 2-nitrobenzoic acid, 3-nitrobenzoic acid, 4-nitrobenzoic acid from** (Ribeiro Da Silva et al., 1999)**, 4-methyl-3-nitrobenzoic acid, 3-methyl-4-nitrobenzoic acid from** (Monte et al., 2001)) **- black diamond for literature data for previous KEMS work (3-nitrophenol, 4-nitrophenol from** (Bannan et al., 2017)**, 4-methyl-2-nitrophenol, 4-methyl-3-nitrophenol, 3-methyl-4-nitrophenol from** (Dang et al., 2019)) **Error bars on the Experimental data points are +/- 1 standard deviation. Section (a) contains nitrophenols, Section (b) contains nitrobenzaldehydes, and Section (c) contains nitrobenzoic acids.**





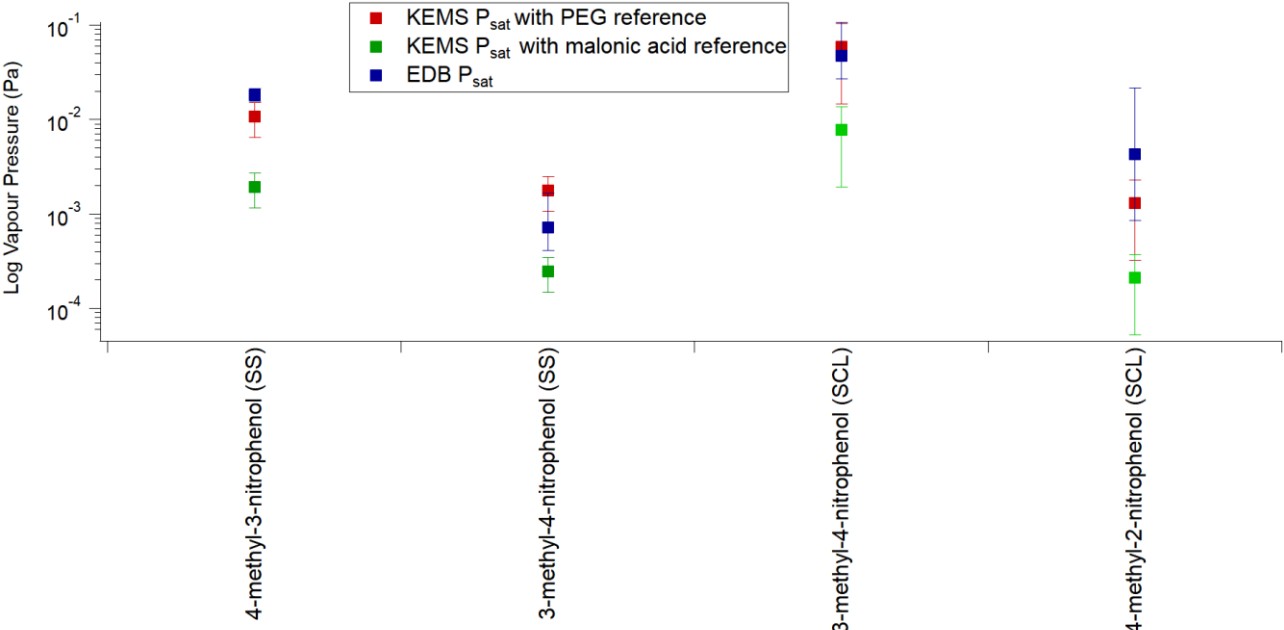

**Figure 9: Comparison of $P^{sat}$ between the EDB and the KEMS using both PEGs and Malonic acid as the reference compound (SS – solid state, SCL – sub-cooled liquid)**

**Table 1: Nitrophenols measured with the KEMS**

| Compound | Structure | CAS | Supplier |
|---|---|---|---|
| 2-nitrophenol | | 88-75-5 | Acros Organics |
| 3-methyl-2-nitrophenol | | 4920-77-8 | Sigma Aldrich |



| 4-methyl-2-nitrophenol | | 119-33-5 | Acros Organics |
|---|---|---|---|
| 5-fluoro-2-nitrophenol | | 446-36-6 | Fluorochem |
| 4-amino-2-nitrophenol | | 119-34-6 | Acros Organics |
| 4-methyl-3-nitrophenol | | 2042-14-0 | Sigma Aldrich |
| 4-chloro-3-nitrophenol | | 610-78-6 | Alfa Aesar |
| 3-methyl-4-nitrophenol | | 2581-34-2 | Fluorochem |



| Compound | Structure | CAS | Supplier |
|---|---|---|---|
| 2-fluoro-4-nitrophenol | | 403-19-0 | Fluorochem |
| 3-fluoro-4-nitrophenol | | 394-41-2 | Acros Organics |

**Table 2: Nitrobenzaldehydes measured with the KEMS**

| Compound | Structure | CAS | Supplier |
|---|---|---|---|
| 2-nitrobenzaldehyde | | 552-89-6 | Sigma Aldrich |
| 3-nitrobenzaldehyde | | 99-61-6 | Sigma Aldrich |
| 2-chloro-5-nitrobenzaldehyde | | 6361-21-3 | Acros Organics |





| | | | |
|---|---|---|---|
| 4-nitrobenzaldehyde | | 555-16-8 | Sigma Aldrich |


**Table 3: Nitrobenzoic acids measured with the KEMS**

| Compound | Structure | CAS | Supplier |
|---|---|---|---|
| 5-chloro-2-nitrobenzoic acid | | 2516-95-2 | Sigma Aldrich |
| 3-nitrobenzoic acid | | 121-92-6 | Sigma Aldrich |
| 4-methyl-3-nitrobenzoic acid | | 96-98-0 | Sigma Aldrich |
| 2-chloro-3-nitrobenzoic acid | | 3970-35-2 | Sigma Aldrich |



| 2-hydroxy-5-nitrobenzoic acid | | 96-97-9 | Sigma Aldrich |
|---|---|---|---|
| 3-methyl-4-nitrobenzoic acid | | 3113-71-1 | Sigma Aldrich |

**Table 4: partial charge of phenolic carbon compared to the H-bond energy ($E_{HB}$)**

| Compound | Partial charge of the phenolic carbon | Remko and Polcin (Remko and Polcin, 1977) $E_{HB}$ (kJ mol$^{-1}$) | Stymne et al. (Stymne et al., 1973) $E_{HB}$ (kJ mol$^{-1}$) |
|---|---|---|---|
| 4-methoxyphenol | 0.222 | 43.5 | 23.8 |
| 2-methoxyphenol | 0.199 | 44.1 | |
| Phenol | 0.294 | 44.7 | 24.3 |
| 3-methoxyphenol | 0.354 | 45.6 | 26.4 |


**Table 5: $P_S^{sat}$ at 298 K, and enthalpies and entropies of sublimation of nitrophenols determined using KEMS**

| Compound | $P_{298}$ (Pa) | $\Delta H_{sub}$ (kJ mol$^{-1}$) | $\Delta S_{sub}$ (J mol$^{-1}$ K$^{-1}$) |
|---|---|---|---|
| 2-nitrophenol | 8.94E-04 | 79.32 | 206.78 |
| 3-methyl-2-nitrophenol | 9.90E-03 | 94.79 | 279.50 |
| 4-methyl-2-nitrophenol | 3.11E-03 | 95.26 | 271.45 |
| 5-fluoro-2-nitrophenol | 4.25E-03 | 95.84 | 276.14 |
| 4-amino-2-nitrophenol | 3.36E-03 | 111.24 | 325.81 |
| 4-methyl-3-nitrophenol | 1.08E-02 | 96.14 | 284.98 |





| | | | |
|---|---|---|---|
| 4-chloro-3-nitrophenol | 2.26E-03 | 104.49 | 299.83 |
| 3-methyl-4-nitrophenol | 1.78E-03 | 90.85 | 251.97 |
| 2-fluoro-4-nitrophenol | 2.75E-02 | 103.76 | 317.90 |
| 3-fluoro-4-nitrophenol | 4.55E-03 | 108.61 | 319.55 |

**Table 6: $P_S^{sat}$ at 298 K, and enthalpies and entropies of sublimation of nitrobenzaldehydes determined using KEMS**

| Compound | $P_{298}$ (Pa) | $\Delta H_{sub}$ (kJ mol$^{-1}$) | $\Delta S_{sub}$ (J mol$^{-1}$ K$^{-1}$) |
|---|---|---|---|
| 2-nitrobenzaldehyde | 3.32E-01 | 73.81 | 238.13 |
| 3-nitrobenzaldehyde | 1.21E-01 | 83.51 | 262.67 |
| 2-chloro-5-nitrobenzaldehyde | 4.21E-02 | 101.26 | 313.39 |
| 4-nitrobenzaldehyde | 3.40E-02 | 103.80 | 320.10 |

**Table 7: $P_S^{sat}$ at 298 K, and enthalpies and entropies of sublimation of nitrobenzoic acids determined using KEMS**

| Compound | $P_{298}$ (Pa) | $\Delta H_{sub}$ (kJ mol$^{-1}$) | $\Delta S_{sub}$ (J mol$^{-1}$ K$^{-1}$) |
|---|---|---|---|
| 5-chloro-2-nitrobenzoic acid | 2.98E-03 | 80.66 | 221.09 |
| 3-nitrobenzoic acid | 1.10E-03 | 87.82 | 237.49 |
| 4-methyl-3-nitrobenzoic acid | 4.67E-03 | 74.66 | 205.82 |
| 2-chloro-3-nitrobenzoic acid | 1.97E-03 | 73.54 | 194.48 |
| 2-hydroxy-5-nitrobenzoic acid | 1.79E-03 | 78.20 | 209.30 |
| 3-methyl-4-nitrobenzoic acid | 3.97E-03 | 65.95 | 175.21 |

**Table 8: $P_L^{sat}$, melting point, and the enthalpy and entropy of fusion of the nitrophenols.**

| Compound | $P_{298}$ (Pa) | $T_m$ (K) | $\Delta H_{fus}$ (kJ mol$^{-1}$) | $\Delta S_{fus}$ (J mol$^{-1}$ K$^{-1}$) |
|---|---|---|---|---|
| 2-nitrophenol | 1.38E-03 | 319.77 | 18.55 | 58.02 |
| 3-methyl-2-nitrophenol | 1.22E-02 | 313.47 | 10.73 | 34.23 |
| 4-methyl-2-nitrophenol | 3.29E-03 | 306.67 | 2.43 | 7.92 |
| 5-fluoro-2-nitrophenol | 5.01E-03 | 309.16 | 11.63 | 37.62 |
| 4-amino-2-nitrophenol | 9.29E-03 | 401.89 | 37.15 | 92.44 |
| 4-methyl-3-nitrophenol | 6.85E-02 | 351.59 | 32.74 | 93.13 |



| 4-chloro-3-nitrophenol | 5.80E-02 | 400.32 | 36.15 | 90.31 |
| 3-methyl-4-nitrophenol | 5.86E-02 | 401.27 | 38.87 | 96.86 |
| 2-fluoro-4-nitrophenol | 6.42E-02 | 394.17 | 9.95 | 25.24 |
| 3-fluoro-4-nitrophenol | 3.32E-02 | 366.46 | 29.36 | 80.12 |

**Table 9: $P_L^{sat}$, melting point, and the enthalpy and entropy of fusion of the nitrobenzaldehydes.**

| Compound | $P_{298}$ (Pa) | $T_m$ (K) | $\Delta H_{fus}$ (kJ mol$^{-1}$) | $\Delta S_{fus}$ (J mol$^{-1}$ K$^{-1}$) |
|---|---|---|---|---|
| 2-nitrobenzaldehyde | 2.15E+00 | 317.66 | 77.98 | 245.49 |
| 3-nitrobenzaldehyde | 2.75E-01 | 332.71 | 20.66 | 62.09 |
| 2-chloro-5-nitrobenzaldehyde | 8.41E-02 | 353.38 | 12.30 | 34.82 |
| 4-nitrobenzaldehyde | 1.93E-01 | 380.40 | 22.51 | 59.16 |


**Table 10: $P_L^{sat}$, melting point, and the enthalpy and entropy of fusion of the nitrobenzoic acids.**

| Compound | $P_{298}$ (Pa) | $T_m$ (K) | $\Delta H_{fus}$ (kJ mol$^{-1}$) | $\Delta S_{fus}$ (J mol$^{-1}$ K$^{-1}$) |
|---|---|---|---|---|
| 5-chloro-2-nitrobenzoic acid | 1.40E-02 | 458.17 | 13.75 | 30.00 |
| 3-nitrobenzoic acid | 1.90E-03 | 418.03 | 5.57 | 13.33 |
| 4-methyl-3-nitrobenzoic acid | 5.76E-02 | 464.70 | 21.87 | 47.06 |
| 2-chloro-3-nitrobenzoic acid | 6.29E-03 | 458.17 | 10.28 | 22.43 |
| 2-hydroxy-5-nitrobenzoic acid | 1.87E-02 | 505.55 | 18.68 | 36.95 |
| 3-methyl-4-nitrobenzoic acid | 3.04E-01 | 492.43 | 35.39 | 71.86 |

**Table 11: Comparison between nitrophenols measured in this paper and by Dang et al. (2019)**

| Compound | Solid State $P_{298}$ (Pa) | Sub-Cooled $P_{298}$ (Pa) | |
|---|---|---|---|
| 4-methyl-3-nitrophenol | 1.08 ± 0.43E-02 | 6.85 ± 5.14E-02 | This work - PEG reference |
| | 1.94 ± 0.78E-03 | 1.23 ± 0.92E-02 | This work - malonic |



| | | | |
|---|---|---|---|
| | | | acid reference |
| | $2.46 \pm 0.98$E-03 | $4.85 \pm 3.64$E-03 | Dang et al. (Dang et al., 2019) |
| | $1.84^{+0.30}_{-0.27}$E-02 | | EDB |
| 3-methyl-4-nitrophenol | $1.78 \pm 0.71$E-03 | $5.86 \pm 4.40$E-02 | This work - PEG reference |
| | $2.45 \pm 0.98$E-04 | $7.80 \pm 5.85$E-03 | This work - malonic acid reference |
| | $2.28 \pm 0.91$E-04 | $3.78 \pm 2.84$E-03 | Dang et al. (Dang et al., 2019) |
| | $7.20^{+9.30}_{-3.10}$E-04 | $4.70^{+6.00}_{-2.00}$E-02 | EDB |
| 4-methyl-2-nitrophenol | $3.11 \pm 1.24$E-03 | $3.29 \pm 2.47$E-03 | This work - PEG reference |
| | $5.61 \pm 2.24$E-04 | $5.76 \pm 4.32$E-04 | This work - malonic acid reference |
| | $5.72 \pm 2.29$E-04 | $5.97 \pm 4.48$E-04 | Dang et al. (Dang et al., 2019) |

**Table 12: Comparison between the $P^{sat}$ for 4-methyl-2-nitrophenol measured with the KEMS and estimated based on behaviour within the EDB**

| | $P_L^{sat}$ (Pa) | |
|---|---|---|
| 4-methyl-2-nitrophenol | $1.30 \pm 0.98$E-03 | KEMS with PEG reference |
| | $2.10 \pm 1.57$E-04 | KEMS with malonic acid reference |
| | $4.30^{+17.20}_{-3.44}$E-03 | EDB estimation based on Eq. 3 |