# Peer review of "Measured solid state and sub-cooled liquid vapour pressures of nitroaromatics using Knudsen effusion mass spectrometry"

_Atmospheric Chemistry and Physics, 2020_

## Referee Comment (RC1) · Anonymous Referee #1 · 4 Mar 2020

Organic nitrates in general and aromatic organic nitrates are important atmospheric compounds that contribute to SOA formation. From this point of view it is important to have saturation vapor pressures and suited prediction methods at hand, that will aid mechanistic understanding and modelling of SOA formation from this class of compounds.

It is known, that GCM can strongly overestimate the vapor pressures of organic nitrates and of multiple functionalized organic molecules, in general. One reason is that the GCM are based on not-suited or too small training sets. From this point of view the manuscript is absolutely timely and it addresses an important aspect in atmospheric

physical chemistry.

Essentially, the authors provide vapor pressures and thermodynamic data for 20 functionalized aromatic nitrates, grouped into Nitro-phenols, Nitro-benzaldehydes, Nitro-benzoic acids. The measured vapor pressures measured over the solid compounds were converted to vapor pressure over subcooled liquids for more general use.

In addition the authors investigated the role of the calibration standards in their KEMS and corroborated their findings by comparison to selected EDB results. The authors try to systematize their results in concepts of inductive and mesomeric effects of aromatic substituents and in terms of H bond donor strength or dipole moments. The material presented in the manuscript is very good and useful. However, I identify some weakness in the presentation of the material. Especially, the result section 4.1 is difficult to follow and it is mixing results and discussion. Related to that, I also suggest to support the textual description by more graphics/diagrams (see comments below).

The manuscript should be published in ACP, but needs major revisions along the lines below.

Major comments

Why do you consider the comparison to GCM EVAPORATION at all, if it should not be used for aromatic compounds? I suggest to leave out the parts regarding EVAPORATION.

The Results section contains results and discussions. Either results and discussions must be separated into two independent section. Or the type of section should be indicated by "Result and Discussion".

Line 243-260: Why do you put so much emphasize on the methoxy phenols? This has not much to do with your work and the concepts of inductive, mesomeric and H-bond effects are so general that you don't have to introduce it by this specific example. In any case, it is not your result and therefore misplaced in a Result section. Moreover, I don't

understand what is supposed to be learned from Figure 3, it is not showing the overall importance of the H-bond (line 244). The methoxy phenols could help the discussion of your findings, though, if you could relate their vapor pressures to their H-bond abilities.

I suggest to omitting whole part with the methoxy phenols and Figure 3 and Table 4 should be skipped (In Table 4, something happened to the table header). That would make some space for my next suggestion to put more illustrations to section 4.1.

line 261-312: Here you present and discuss your findings for the nitro phenols. It is very difficult to follow your description and interpretations based only on the text and on the tables, because you present many numbers in combination with similar looking compound names.

I understand that the authors have access to MOPAC2016 and were able to calculate by themselves the partial charges on the carbon which carries the phenol group and estimate H bond strength as well as calculate dipole moments. I suggest to present all used helping quantities, i.e. (relative) strength of I- and M-effect, partial charge on phenolic C, dipole moments, in an extra table or add it to the Table 5. The authors then should try to plot the vapor pressures as function of (some of) these quantities in addition to table(s) and text.

I know, it may be challenging to clearly arrange that information in a graphical way. However, it would help the readability of the manuscript a lot. E.g. "outliers" could be used as start for your discussion of secondary effects like steric effects, or intramolecular H-bonding (as presented in the current text).

line 313-326: In the same sense as before: do you have dipole moments of the nitrobenzaldhydes? Could you add this information to Table 6, make a graphics and discuss the results in similar terms as the phenols?

line 327-329: Why don't you show that relation in a plot.

line 330-348: In the similar sense as commented above: try to find a good graphical

presentation of your findings. Using partial charge on the carboxylic C, would that enable comparison of the acids to the phenols, in terms of H-bond donor strength?

line 349-362: Summary, yes, it this very informative. I argue again, it would be great to have the proposed diagrams in the previous sections, which show the trends and the exceptions, highlighting the statements in this summary.

Minor comments

line 114 – line 118: I understand that you only used PEG-3 and PEG-4 to calibrate your KEMS? I feel, the discussion of the PEG series is distracting and confusing (me) here. It is covered by the Krieger et al. (2018) reference. If you feel the need to discuss PEG in such detail, I suggest to move it to the supplement.

line 118f: You mentioned the PEG-4 is a suited standard, but you obviously used also PEG-3. What is the quality of the KEMS for PEG-3 measurements?

line 268f: I suggest to take the sentence to the previous paragraph and make the new paragraph after the sentence.

Section 4.5: I suggest to move some details of the EDB measurements to the EDB section 2.3 and to focus here on the comparison itself.

line 560: I would suggest to slightly reformulate. "in non-protic systems the dipole moment. . ."

Figure 9: I think there is space to show all data discussed and given in Tables 11 and 12, also some are less complete.

In general: check your literature input: e.g. McFiggans, O'Meara

If you quote authors directly, the format should be: "As mentioned by Bilde et al. (2015) . . ."

---

## Referee Comment (RC2) · Anonymous Referee #2 · 14 Apr 2020

Saturation vapor pressure (Psat) is essential for understanding and modeling of OA formation. Due the complexity of ambient OA, it is not feasible to measure the Psat and GCMs are frequently used for predicting Psat. However, GCMs can strongly overestimate the Psat due to a range of factors (e.g. underrepresentation of long-chain hydrocarbons and specific functional groups, a lack of data for the impact of intramolecular bonding). Although aromatic nitrates are important atmospheric compounds that contribute to OA formation, quality Psat data for them are not available. In this study, the authors provide Psat and thermodynamic data for 20 functionalized aromatic nitrates, including Nitro-phenols, Nitro-benzaldehydes, Nitrobenzoic acids. They performed a systematic investigation on how H bond effect, steric effects, dipole moments, and

crystallographic packing density affect the Psat. Moreover, the performance of GCMs is evaluated using the measured data set. The data presented in the manuscript are useful and give an insight into the impact of functional group positioning which is missing, or only available in a limited capacity, for the currently available GCMs. While this study can be a good contribution to atmospheric science, the presentation of the results can be improved. Thus, I recommend a major revision. Major Comments 1. Line 224-260 First of all, the authors spend a lot of effort in introducing some general information about inductive, resonance, and H-bond effects. I don't think it is appropriate to put all of this background information in the Results section. This information could be moved to the Theory section or even supporting information. Second, methoxy-phenols are not compounds of interest measured by this study. This study already included a lot of chemical compounds. I believe the authors can use the studied species to illustrate the relationship between H-bond energy and partial charge of the phenolic carbon. Moreover, figure 3 does not contain much useful information, and table 4 should be changed accordingly. 2. Line 270-348: Here, the authors present a lot of results to illustrate how the H-bond effect, steric effects, dipole moments, and crystallographic packing densities affect the Psat. It is challenging to follow the description and interpretations based only on the text. I think a summary table including key parameters involved (e.g. partial charge on phenolic C, dipole moments, crystallographic packing densities) will be beneficial. Additionally, some correlation figures (e.g. partial charge vs Psat, dipole moment vs Psat, crystallographic packing vs Psat) or visual images could be useful for the discussion and for readers to follow. 3. The author evaluated the Psat data predicted by the GCM comprehensively. Could the authors make a summary table to show the features of each GCM method, the performance of the prediction (the difference as compared to measurements), and short explanation of why the predictions differ from measurements. Furthermore, could the authors make a summary to say which prediction method may provide best result for a type of compound? This will help the researchers to get a more reasonable result when use GCMs doe predicting Psat for new compounds. Specific comments 1. Line 37-38. The sentence regarding

SOA formation mechanism is not rigorous. Gas phase photochemical reactions do not produce SOA directly. Another step of gas-to-particle conversion is needed. Moreover, 2. Line 112-123: The discussion on the PEG has been presented by Krieger et al. (2018) already. It is not necessary to show it here again. Moreover, the author stated that "KEMS was able to determine Psat of PEG-4 to PEG-7, through good agreement with the other techniques". Why the author used PEG3 here for calibration if only measurements for PEG-4 to PEG-7 have good agreement? 3. Line 124-125 This sentence seems to be redundant. 4. Line 214-217 Why the measurement temperature range needs to be listed here and why only listed for 5 compounds? 5. Line 268-269: I suggest to take the first sentence to the previous paragraph. 6. Line 381-382. Why the authors still used EVAPORATION to estimate the Psat of studied compounds and used SIMPLO for fluoro-aromatics? It is stated clearly that "A common misuse of GCMs occurs when a GCM is applied to a compound containing functionality not included in the training set, e.g. using EVAPORATION (Compernolle et al., 2011) with aromatic compounds or using SIMPOL (Pankow and Asher, 2008) with compounds containing halogens." (lines 194-196) 7. Line 422, A full stop is needed after "by Dang et al. (2019)" 8. Section 4.5: Details of EDB measurements regarding physical state determination and Psat estimation should be moved to section 2.3. 9. section title "Result" should be replaced by "Result and Discussion". 10. The reference style should be checked throughout. For example, Line 51-52 "Barley and McFiggans (Barley and McFiggans, 2010) and O'Meara et al. (O 'meara et al., 2014)" should be changed to "Barley and McFiggans (2010) and O'Meara et al. (2014)". 11. There is not much information in table 12. These numbers are listed in the text and displayed in Figure 9 already. 12. PL sat sometimes are in Bold in the text (e.g. on Page 17). 13. I think table 5,6,7 can be merged into 1 table, also table 8,9,10. These two sets of tables show similar information. 14. The quality of Figure 4 is poor.

---

## Author Comment (AC1) · 26 May 2020

Response to comments on:

Measured solid state and sub-cooled liquid vapour pressures of nitroaromatics using Knudsen effusion mass spectrometry

We thank the reviewers and editor for their time evaluating this manuscript and their comments relating to this work. The corrections and additions made as a result of

these comments have greatly improved the focus of this work.

The response to each of the referees' points, which are repeated in black, is provided in blue text with the new additions to the text in the paper in red. References to the original text are made in orange.

Anonymous Referee #1:

Major comment 1

Why do you consider the comparison to GCM EVAPORATION at all, if it should not be used for aromatic compounds? I suggest to leave out the parts regarding EVAPORATION.

The authors agree with this comment. EVAPORATION is a commonly used GCM and it is important to show where it is not appropriate to use it. Additional text had been added to the introduction discussing the strengths of EVAPORATION, before explaining why it is not suited to aromatic compounds and its omission from the comparisons made in this study.

(Line 59 – 62 All Markup updated manuscript):

For example, in the assessment by O'Meara et al. (2014), for the compounds to which it is applicable, EVAPORATION (Compernolle et al., 2011) was found to give the minimum mean absolute error, the highest accuracy for SOA loading estimates and the highest accuracy for SOA composition. Despite this EVAPORATION should not be used for aromatic compounds, as there are no aromatic compounds in the
parametrisation dataset (Compernolle et al., 2011).

Mention of EVAPORATION has been removed from the results and discussion section, and the EVAPORATION data points have been removed from Figure 10 in the revised manuscript (Figure 8 in the original manuscript).

Major comment 2

The Results section contains results and discussions. Either results and discussions must be separated into two independent section. Or the type of section should be indicated by "Result and Discussion".

Agreed, as a result we have changed the name of the section to "Results and Discussion" as suggested.

Major comment 3

Line 243-260: Why do you put so much emphasize on the methoxy phenols? This has not much to do with your work and the concepts of inductive, mesomeric and H-bond effects are so general that you don't have to introduce it by this specific example. In any case, it is not your result and therefore misplaced in a Result section. Moreover, I don't understand what is supposed to be learned from Figure 3, it is not showing the overall importance of the H-bond (line 244). The methoxy phenols could help the discussion of your findings, though, if you could relate their vapor pressures to their H-bond abilities. I suggest to omitting whole part with the methoxy phenols and Figure 3 and Table 4 should be skipped (In Table 4, something happened to the table

header). That would make some space for my next suggestion to put more illustrations to section 4.1.

Discussion of methoxyphenols has been removed along with Figure 3 and Table 4 as suggested. The more general resonance and inductive effect theory has been moved the theory section in subsection 3.3 titled Inductive and resonance effects.

Major comment 4

line 261-312: Here you present and discuss your findings for the nitro phenols. It is very difficult to follow your description and interpretations based only on the text and on the tables, because you present many numbers in combination with similar looking compound names. I understand that the authors have access to MOPAC2016 and were able to calculate by themselves the partial charges on the carbon which carries the phenol group and estimate H bond strength as well as calculate dipole moments. I suggest to present all used helping quantities, i.e. (relative) strength of I- and M-effect, partial charge on phenolic C, dipole moments, in an extra table or add it to the Table 5. The authors then should try to plot the vapor pressures as function of (some of) these quantities in addition to table(s) and text. I know, it may be challenging to clearly arrange that information in a graphical way. However, it would help the readability of the manuscript a lot. E.g. "outliers" could be used as start for your discussion of secondary effects like steric effects, or intramolecular H-bonding (as presented in the current text).

The partial charges of the phenolic carbons have been added to table 4, along with a new figure (Figure 5 in the updated manuscript) as requested. Figure 5 shows a plot of $P_S^{sat}$ vs partial charge of the phenolic carbon with the methyl nitrophenols plotted in blue, fluoro nitrophenols plotted in red, and the other/outlier nitrophenols plotted

in green with each compound labelled. This together with explicit references to the Figure 5 in the text when directly comparing the $P_S^{sat}$ of two compounds should help the readability of the section significantly. By referencing figure 5 additional discussion of the compounds marked in green has been added to explain why they deviate from what would be expected given one factor or another.

See Figure 5 in the supplement to this Author comment

(Line 320 – 321 All Markup updated manuscript):

A plot of $P_S^{sat}$ vs the partial charge of the phenolic carbon for the nitrophenols can be found in Fig. 5.

(Line 329 All Markup updated manuscript):

For the fluoro nitrophenols, as shown in Fig. 5, as the partial charge of the phenolic carbon increases the $P_S^{sat}$ increases.

(Line 330 – 332 All Markup updated manuscript):

A similar trend occurs in the methyl nitrophenols as in the fluoro nitrophenols with a larger partial charge of the phenolic carbon corresponding to a lower $P_S^{sat}$, as shown in Fig 5. 3-methyl-2-nitrophenol is an exception to this and is discussed shortly.

(Line 335 – 337 All Markup updated manuscript):

[Figure]

As shown in Fig. 5 3-methyl-2-nitrophenol would be expected to have a much lower $P_S^{sat}$ than is observed due to the high partial charge on the phenolic carbon.

(Line 341 – 345 All Markup updated manuscript):

Whilst 3-methyl-2-nitrophenol has a higher $P_S^{sat}$ than is expected given the partial charge on the phenolic carbon, 4-amino-2-nitrophenol has a much lower $P_S^{sat}$ (Fig. 5). This is likely due to 4-amino-2-nitrophenol being capable of forming more than one hydrogen bond, whereas all the other compounds investigated were only capable of forming one H-bond. However, despite 4-amino-2-nitrophenol being capable of forming more than 1 H-bond, replacing the methyl group on 4-methyl-2-nitrophenol with an amino group to form 4-amino-2-nitrophenol surprisingly increases the $P_S^{sat}$ from 3.11E-03 Pa to 3.36E-03 Pa.

(Line 351 – 355 All Markup updated manuscript):

4-amino-2-nitrophenol is a good example of a compound with multiple competing factors affecting $P_S^{sat}$ leading to higher $P_S^{sat}$ than would be expected due to one factor and lower $P_S^{sat}$ than expected from another.

Similar to 4-amino-2-nitrophenol, 4-chloro-3-nitrophenol also has a lower $P_S^{sat}$ than expected according to the partial charge of the phenolic carbon. This can be seen in Fig. 5. Unlike 4-amino-2-nitrophenol the explanation for 4-chloro-3-nitrophenol is simpler.

(Line 364 – 367 All Markup updated manuscript):

[Figure]

It is possible that the crystallographic packing density of 3-methyl-4-nitrophenol is higher although no data is available to support this, although when looking at $P_L^{sat}$ data (Section 4.2) 3-methyl-4-nitrophenol exhibits a higher $P_L^{sat}$ than 3-fluoro-4-nitrophenol which is what would be expected given the respective partial charges of the phenolic carbons.

Replaced mention of dipole moment between line 358 and 361 with partial charge as on looking at the data, as a whole dipole moments have very poor correlation with vapour pressure, and for H-bonding compounds partial charge and strength of H-bonds will be much more important.

(Line 359 – 362 All Markup updated manuscript):

The methyl group cannot engage in intermolecular H-bonding, it will sterically hinder any H-bonding that the NO2 group undergoes and it reduces the partial charge of the phenolic carbon of the molecule (from 0.379 to 0.362) (Stewart, 2016) which would reduce the strength of H-bonding interactions between the molecules.

Removed mention of 3-methyl-2-nitrophenol between line 368 and 370 as it is clearly an outlier.

Major comment 5

line 313-326: In the same sense as before: do you have dipole moments of the nitrobenzaldhydes? Could you add this information to Table 6, make a graphics and

discuss the results in similar terms as the phenols?

We agree that this is a useful addition. We have therefore added column to Table 5 containing the crystallographic packing density of the nitrobenzaldehydes and added a new figure (Figure 7) showing a plot of $P_S^{sat}$ vs Packing density. Figure 7 shows a very strong correlation between $P_S^{sat}$ vs Packing density for the nitrobenzaldehydes.

See Figure 7 in the supplement to this Author comment

Major comment 6

line 327-329: Why don't you show that relation in a plot

This has been removed as it is more of an observation than anything else and does not add very much to the discussion.

Major comment 7

line 330-348: In the similar sense as commented above: try to find a good graph-ical presentation of your findings. Using partial charge on the carboxylic C, would that enable comparison of the acids to the phenols, in terms of H-bond donor strength?

The partial charge of the carboxylic carbon has been added to Table 7, and a new figure has been made (Figure 8). Figure 8 contains both the nitrophenols and nitrobenzoic acids, with a zoomed in section of the nitrobenzoic acids. Figure 8 plots $P_S^{sat}$ vs partial charge of the phenolic/carboxylic carbon. Each individual carboxylic

acid is labelled. In this section originally there were some errors that have been corrected. A more detailed comparison of the nitrophenols and nitrobenzoic acids has been added. Figure 9 (originally Figure 7) has been adjusted.

See Figure 8 in the supplement to this Author comment

See Figure 9 in the supplement to this Author comment

Corrected error between lines 390 and 392.

(Line 390 – 392 All Markup updated manuscript):

Its isomer 3-methyl-4-nitrobenzoic acid possesses a slightly lower $P_S^{sat}$ (3.97E-03) as well as a slightly lower partial charge of the carboxylic carbon (0.644 vs 0.628) although the difference in $P_S^{sat}$ is not significant.

Removed incorrect comparison between 3-methyl-4-nitrophenol, 4-methyl-3-nitrophenol, 3-methyl-4-nitrobenzoic acid and 4-methyl-3-nitrobenzoic acid between lines 391 and 396.

Added more detailed comparison between nitrophenols and nitrobenzoic acids as a whole between lines 408 to 418.

(Line 408 – 418 All Markup updated manuscript):

[Figure]

When comparing nitrobenzoic acids as a whole with nitrophenols, nitrobenzoic acids have a much higher $P_S^{sat}$ than would be expected based solely on the partial charges of the carboxylic carbon. As can be seen in Fig. 8, there is overlap in the range of $P_S^{sat}$ for the nitrobenzoic acids and many of the nitrophenols, however there is no overlap in terms of partial charges of the carboxylic and phenolic carbons, with all of the nitrobenzoic acids having partial charges of the carboxylic carbon greater than 0.6, whilst the nitrophenols had much lower partial charges of the phenolic carbon between 0.2 and 0.4. It is widely known that the H-bonds of carboxylic acids are stronger than the H-bonds of alcohols (Ouellette et al., 2015b) so therefore it would be expected that the carboxylic acids would have a lower $P_S^{sat}$. A likely reason as to why the $P_S^{sat}$ of the nitrobenzoic acids is higher than would be expected, compared to the nitrophenols, based only on the partial charge of the carboxylic carbon is the propensity for carboxylic acids to dimerise (see Fig. 9). Nitrophenols are unable to dimerise, instead being able to form H-bonds with up to 2 other molecules as shown in Fig. 9. By dimerising the nitrobenzoic acids, despite having much stronger H-bonds than the nitrophenols, will not have a proportionally lower $P_S^{sat}$.

Major comment 8

line 349-362: Summary, yes, it this very informative. I argue again, it would be great to have the proposed diagrams in the previous sections, which show the trends and the exceptions, highlighting the statements in this summary

New figures have been added to the previous sections to more clearly show the properties that have a large impact on $P_S^{sat}$. Mention of dipole moments have been removed from this section with a sentence added at the end stating that dipole moments showed little impact on $P_S^{sat}$, with dipole moments showing positive correlation, negative correlation and no correlation with $P_S^{sat}$.

(Line 433 All Markup updated manuscript):

Dipole moments were also investigated but overall showed very little impact on $P_S^{sat}$.

Minor comment 1

line 114 – line 118: I understand that you only used PEG-3 and PEG-4 to calibrate your KEMS? I feel, the discussion of the PEG series is distracting and confusing (me) here. It is covered by the Krieger et al. (2018) reference. If you feel the need to discuss PEG in such detail, I suggest to move it to the supplement.

The authors agree that this is unnecessary detail covered by the Krieger et al. (2018) study and this additional discussion has therefore been removed.

Minor comment 2

line 118f: You mentioned the PEG-4 is a suited standard, but you obviously used also PEG-3. What is the quality of the KEMS for PEG-3 measurements?

The KEMS wasn't used to measure PEG-3 in Krieger et al. (2018), but it is a suitable reference standard and over the multiple measurements taken during data collection PEG-3 agreed within 20 - 30 % of the experimental $P_S^{sat}$ measurements from Krieger et al. (2018). Information about the quality of PEG-3 measurements using the KEMS has been added.

[Figure]

(Line 115-120 All Markup updated manuscript):

The KEMS has been shown to accurately measure the $P^{sat}$ of PEG-4 in the study by Krieger et al. (2018) but the KEMS did not measure the $P^{sat}$ of PEG-3. In this study when using PEG-4 as a reference compound for PEG-3 the measured $P^{sat}$ of PEG-3 had an error of 30 % compared to the experimental values from Krieger et al. ( 2018), well within the quoted 40 % error margin of the KEMS (Booth et al., 2009). When using PEG-3 as the reference compound for PEG-4 the measured $P^{sat}$ of PEG-4 had an error of 20 %.

Minor comment 3

line 268f: I suggest to take the sentence to the previous paragraph and make the new paragraph after the sentence.

This has been done as suggested.

Minor comment 4

Section 4.5: I suggest to move some details of the EDB measurements to the EDB section 2.3 and to focus here on the comparison itself.

This has been done. The following has been moved from section 4.5 to 2.3.

(Line 161 – 165 All Markup updated manuscript):

The recently published paper by Dang et al. (2019) measured the $P^{sat}$ of several of the same compounds that are studied in this paper using the same KEMS system, however in this study the newly defined best practice reference sample was used (Krieger et al., 2018), whereas Dang et al. (2019) used malonic acid. The difference in reference compound led to a discrepancy in the experimental $P^{sat}$. Supporting measurements for the compounds were performed using the EDB from ETH Zurich in order to rule out instrumental problem with the KEMS.

(Line 175 – 192 All Markup updated manuscript):

As single particles injected from a dilute solution may either stay in a supersaturated, liquid state or crystallize, it is important to identify its physical state. For 4-methyl-3-nitrophenol a 3 % solution dissolved in isopropanol was injected into the EDB. After the injection and fast evaporation of the isopropanol, all particles were non-spherical, but with only small deviations from a sphere, meaning that it was unclear whether the phase was amorphous or crystalline. To determine the phase of this first experiment, a second experiment was performed, where a solid particle was injected directly into the EDB. Mass loss with time was measured by following the DC voltage necessary to compensate the gravitational force acting on the particle to keep the particle levitating. When comparing the $P^{sat}$ from both of these experiments it is clear that the initial measurement of 4-methyl-3-nitrophenol was in the crystalline phase. 3-methyl-4-nitrophenol was only injected as a solution but the particle crystallized and was clearly in the solid state. 4-methyl-2-nitrophenol was injected as both a 3 % and 10 % solution. Despite being able to trap a particle, the particle would completely evaporate within about 30 seconds. This evaporation time scale is too small to allow the EDB to collect any quantitative data. Using the equation for large particles neglecting evaporative cooling (Hinds, 1999) (Eq. 2) it is possible to estimate $P^{sat}_L$

$$t = \frac{R\rho \cdot d_p^2}{8DM\frac{P_{sat}}{T}} (2)$$

where t is the time that the particle was trapped within the cell of the EDB, R is the ideal gas constant, is the density of the particle, dp is the diameter of the particle, D is the diffusion coefficient, M is the molecular mass, T is the temperature, and $P^{sat}$ is the saturation vapour pressure. Eq. 2 gives approximately 4.3E-03 Pa for $P_L^{sat}$ at 290 K.

Minor comment 5

line 560: I would suggest to slightly reformulate. "in non-protic systems the dipole moment. . ."

This part of the sentence has now been removed as dipole moments are no longer being considered an important factor for $P_S^{sat}$.

Minor comment 6

Figure 9: I think there is space to show all data discussed and given in Tables 11 and 12, also some are less complete.

The data not included in Figure 9 (now Figure 11) is a second set of measurements using malonic acid as a reference compound. As malonic acid is already represented in the figure including the second set isn't necessary.

Table 12 has been removed at the suggestion of another reviewer.

Anonymous referee 2:

Major comment 1

Line 224- 260 First of all, the authors spend a lot of effort in introducing some general information about inductive, resonance, and H-bond effects. I don't think it is appropriate to put all of this background information in the Results section. This information could be moved to the Theory section or even supporting information. Second, methoxy-phenols are not compounds of interest measured by this study. This study already included a lot of chemical compounds. I believe the authors can use the studied species to illustrate the relationship between H-bond energy and partial charge of the phenolic carbon. Moreover, figure 3 does not contain much useful information, and table 4 should be changed accordingly.

The more general Inductive and resonance theory has been moved to a new sub section of the theory section (section 3.3 Inductive and resonance theory). Discussion of methoxy phenols, as well as figure 3 and table 4 have been removed.

Major Comment 2

Line 270-348: Here, the authors present a lot of results to illustrate how the H-bond effect, steric effects, dipole moments, and crystallographic packing densities affect the Psat. It is challenging to follow the description and interpretations based only on the text. I think a summary table including key parameters involved (e.g. partial charge on phenolic C, dipole moments, crystallographic packing densities) will be beneficial.

Additionally, some correlation figures (e.g. partial charge vs Psat, dipole moment vs Psat, crystallographic packing vs Psat) or visual images could be useful for the discussion and for readers to follow.

Many of the comments here were also raised by reviewer 1. We have therefore made the changes recommended as a priority. The partial charge of the phenolic carbon has been added to Table 4 (originally Table 5), crystallographic packing density has been added to Table 5 (originally Table 6) and partial charge of the carboxylic carbon has been added to Table 6 (originally Table 7). In line with the comments from reviewer 1, Figure 5 has been added showing $P_S^{sat}$ vs partial charge, Figure 7 has been added showing a plot of $P_S^{sat}$ vs Packing density and Figure 8 has been added illustrating $P_S^{sat}$ vs partial charge of the phenolic/carboxylic carbon. Detailed discussions of each figure has now also been added to aid the interpretation of the results presented here. Full details of the changes made between lines 271-349 in the original manuscript are given in the response Anonymous Referee 1 between Major comment 4 to Major comment 7 earlier in this document.

Major comment 3

The author evaluated the Psat data predicted by the GCM comprehensively. Could the authors make a summary table to show the features of each GCM method, the performance of the prediction (the difference as compared to measurements), and short explanation of why the predictions differ from measurements. Furthermore, could the authors make a summary to say which prediction method may provide best result for a type of compound? This will help the researchers to get a more reasonable result when use GCMs doe predicting Psat for new compounds.

We agree that a summary table is of use for ease of interpretation. We have therefore added a summary table (Table 8) containing the average order of magnitude difference between the predicted and measured $P^{sat}$ for nitrophenols, nitrobenzaldehydes, nitrobenzoic acids and all three combined.

[revised manuscript text omitted]

Specific comment 1

Line 37-38. The sentence regarding SOA formation mechanism is not rigorous. Gas phase photochemical reactions do not produce SOA directly. Another step of gas-to-particle conversion is needed.

Sentence has been adjusted.

(Line 37-38 All Markup updated manuscript):

SOA are not emitted into the atmosphere directly as aerosols, but instead form through atmospheric processes such as gas phase photochemical reactions followed by gas-to-particle partitioning in the atmosphere (Pöschl, 2005).

Specific comment 2

Line 112-123: The discussion on the PEG has been presented by Krieger et al. (2018) already. It is not necessary to show it here again. Moreover, the author stated that "KEMS was able to determine Psat of PEG-4 to PEG-7, through good agreement with the other techniques". Why the author used PEG3 here for calibration if only

measurements for PEG-4 to PEG-7 have good agreement?

This point was also raised by reviewer 1. The more general PEG discussion has been removed and a comment on using PEG 3 as a reference compound has been added. The full changes that have been made are shown in response to Anonymous Referee 1 minor comment 1 and minor comment 2.

Specific comment 3

Line 124-125 This sentence seems to be redundant.

Sentence has been removed.

Specific comment 4

Line 214-217 Why the measurement temperature range needs to be listed here and why only listed for 5 compounds?

The 5 compounds with temperature range listed were those that melted during the temperature ramp up to 328 K. Moved the sentences around to make this clearer.

(Line 259 – 262 All Markup updated manuscript):

Measurements were made at increments of 5 K from 298 to 328 K with the exception of the following compounds that melted during the temperature ramp. 2-nitrophenol

was measured between 298 K and 318 K, 3-methyl-4-nitrophenol was measured between 298 K and 313K, 4-methyl-2-nitrophenol was measured between 298 K and 303 K, 5-fluoro-2-nitrophenol was measured between 298 K and 308 K, and 2-nitrobenzaldehyde was measured between 298 K and 313 K

Specific comment 5

Line 268-269: I suggest to take the first sentence to the previous paragraph.

Done as suggested.

Specific comment 6

Line 381-382. Why the authors still used EVAPORATION to estimate the Psat of studied compounds and used SIMPLO for fluoro-aromatics? It is stated clearly that "A common misuse of GCMs occurs when a GCM is applied to a compound containing functionality not included in the training set, e.g. using EVAPORATION (Compernolle et al., 2011) with aromatic compounds or using SIMPOL (Pankow and Asher, 2008) with compounds containing halogens." (lines 194-196)

The use of EVAPORATION was also raised by reviewer 1. EVAPORATION is still discussed in the introduction as it is a commonly used GCM, but has ben omitted from the results and discussion section. EVAPORATION has also been removed from Figure 10 (originally Figure 8). Full details of the changes regarding EVAPOTATION are given in response to Anonymous Referee #1 Major comment 1.

[Figure]

When using SIMPOL for halogenated species, despite on paper not being a suitable GCM, SIMPOL performed the best for all of the halogenated species. For this reason, it has been left in. An additional paragraph has been added to draw attention to this.

(Line 529 – 534 All Markup updated manuscript):

One surprising observation comes when looking at the halogenated nitroaromatics. SIMPOL (Pankow and Asher, 2008) has the smallest order of magnitude difference between experimental and predicted $P_L^{sat}$ for all of the halogenated nitroaromatics in this study. This is particularly surprising as SIMPOL (Pankow and Asher, 2008) contains no halogenated compounds in its fitting data set, whereas the other GCMs do. This implies that accurately predicting the impact on $P_L^{sat}$ of carbon skeleton and other functional groups such as, nitro, hydroxy, aldehyde and carboxylic acid are more important than the impact of a chloro or fluoro group.

Specific comment 7

Line 422, A full stop is needed after "by Dang et al. (2019)"

Full stop added.

Specific comment 8

Section 4.5: Details of EDB measurements regarding physical state determination and Psat estimation should be moved to section 2.3

This point was also raised by reviewer 1. The details of details of the physical state determination and Psat estimation have been moved to section 2.3 as suggested. Full details of the changes made can be found in response to Anonymous Referee 1 minor comment 4.

Specific comment 9

section title "Result" should be replaced by "Result and Discussion".

This has been changed as suggested.

Specific comment 10

The reference style should be checked throughout. For example, Line 51-52 "Barley and McFiggans (Barley and McFiggans, 2010) and O'Meara et al. (O 'meara et al., 2014)" should be changed to "Barley and McFiggans (2010) and O'Meara et al. (2014)".

This has been done.

Specific comment 11

There is not much information in table 12. These numbers are listed in the text and displayed in Figure 9 already

Table 12 has been removed.

Specific comment 12

PL sat sometimes are in Bold in the text (e.g. on Page 17).

Bold removed where present for PL sat.

Specific comment 13

I think table 5,6,7 can be merged into 1 table, also table 8,9,10. These two sets of tables show similar information.

Additional data has been added to tables 5, 6 and 7 (now tables 4, 5, 6) now containing different information. Also, as the data of each is mostly discussed separately, I think separate tables are appropriate. Tables 8, 9 and 10 have been merged.

Specific comment 14

The quality of Figure 4 is poor.

Figure has been remade at a higher quality.

**Supplement:**

[Figure]

**Figure 5: $P_S^{sat}$ vs partial charge of the phenolic carbon of the nitrophenols.**

[Figure]

**Figure 7: $P_S^{sat}$ vs Packing density of the nitrobenzaldehydes**

[Figure]

Figure 8: $P_S^{sat}$ vs partial charge of the phenolic/carboxylic carbon of the nitrophenols and nitrobenzoic acids.

Figure 9: Diagram demonstrating how the a carboxylic acid functionality allows a molecule to **dimerise using H-bonds in 4-methyl-3-nitrobenzoic acid (left)** whilst a hydroxyl group only allows for hydrogen bonding to two other molecules **with no opportunity to dimerise in 4-methyl-3-nitrophenol (right).**

---

## Author Response (AR1)

**Measured solid state and sub-cooled liquid vapour pressures of nitroaromatics using Knudsen effusion mass spectrometry**

Petroc D. Shelley1, Thomas J. Bannan1, Stephen D. Worrall2, M. Rami Alfarra1,3, Ulrich K. Krieger4, Carl J. Percival5, Arthur Garforth6, David Topping1

[revised manuscript text omitted]